# Interaction of soil water and groundwater during the freezing-thawing cycle: field observations and numerical modeling

Hong-Yu Xie[1], Xiao-Wei Jiang[1*], Shu-Cong Tan[1], Li Wan[1], Xu-Sheng Wang[1], Si-Hai Liang[1], Yijian Zeng[2]

1. MOE (Ministry of Education) Key Laboratory of Groundwater Circulation and Evolution, China University of Geosciences, Beijing 100083, China

2. Department of Water Resources, ITC Faculty of Geo-Information Science and Earth Observation, University of Twente, Enschede, the Netherlands

*Correspondence to: Xiao-Wei Jiang (jxw@cugb.edu.cn)

## Abstract

Freezing-induced groundwater level decline are widely observed in regions with shallow water table, but many existing studies on freezing-induced groundwater migration do not account for freezing−induced water level fluctuations. Here, by combining detailed field observations of liquid soil water content and groundwater level fluctuations at a site in the Ordos Plateau, China and numerical modeling, we showed that the interaction of soil water and groundwater dynamics were controlled by wintertime atmospheric conditions and topographically-driven lateral groundwater inflow. With an initial water table depth of 120 cm and a lateral groundwater inflow rate of 1.03 mm/d, the observed freezing and thawing-induced fluctuations of soil water content and groundwater level are well reproduced. By calculating the budget of groundwater, the mean upward flux of freezing-induced groundwater loss is 1.46 mm/d for 93 days, while the mean flux of thawing-induced groundwater recharge is as high as 3.94 mm/d for 32 days. These results could be useful for local water resources management when encountering seasonally frozen soils, and for future studies on two- or three-dimensional transient groundwater flow in semi-arid and seasonally frozen regions. By comparing models under a series of conditions, we found the magnitude of freezing-induced groundwater loss decreases with initial water table depth and increases with the rate of groundwater inflow. We also found a fixed-head lower boundary condition would overestimate freezing-induced groundwater migration when the water table depth is shallow. Therefore, an accurate characterization of freezing-induced water table decline is critical to quantifying the contribution of

groundwater to hydrological and ecological processes in cold regions.

## 1 Introduction

Frozen soils, which can be divided into permafrost and seasonally frozen soils, have great impacts on many hydrological, hydrogeological processes and ecological processes (Nelson, 2003;Kurylyk et al., 2014;Schuur et al., 2015;Walvoord and Kurylyk, 2016;Evans et al., 2018). In seasonally frozen soils, the behaviors of subsurface water flow and storage are distinct from those in colder regions with permafrost as well as in warmer regions without frozen soil (Ireson et al., 2013). Understanding the effects of soil freezing and thawing on subsurface water flow and storage is a necessary path for better water resources management in seasonally frozen regions (Yu et al., 2020).

By blocking pores and reducing hydraulic conductivity, soil freezing could lead to decreased infiltration (van der Kamp et al., 2003;Iwata et al., 2008;Demand et al., 2019) and limited surface evaporation (Kaneko et al., 2006;Wu et al., 2016). As a result of cryosuction generated by soil freezing (Williams and Smith, 1989;Hohmann, 1997;Yu et al., 2018;Zheng et al., 2021), water migrates from the unfrozen zone to the freezing front. By assuming that freezing-induced water migration is restricted within the shallow part of unsaturated zone, many numerical studies deployed a free drainage lower boundary condition and examined the effects of freezing-induced water redistribution on infiltration and runoff (Cherkauer and Lettenmaier, 1999;Okkonen et al., 2017;Zhang et al., 2017b). In fact, when the water table is shallow, groundwater in the saturated zone can also be migrated to the freezing front (Harlan, 1973;Shoop and Bigl, 1997;Stähli et al., 1999;Hansson and Lundin, 2006;Ireson et al., 2013;Chen et al., 2019). Although there are many field observations of freezing-induced groundwater level decline since the 1950s (Drescher, 1955;Schneider, 1961;Vinnikov et al., 1996;Daniel and Staricka, 2000;van der Kamp et al., 2003;Ireson et al., 2013;Zhang et al., 2019), numerical research that satisfactorily considers both freezing-induced water gain in the frozen zone and freezing-induced groundwater level decline in the saturated zone is limited.

Freezing-induced water migration from the saturated zone to the freezing front could strengthen the possibility of frost heave (Chamberlain, 1981;Bronfenbrener and Bronfenbrener, 2010;Rui et al., 2019). During the thawing stage, accumulation of thawed water above the frozen zone would accelerate soil evaporation (Fetzer et al., 2017;Vanderborgh et al., 2017;Li et al., 2020), which is critical to soil salinization (Liu et al., 2009;Lopez et al., 2010;Bechtold et al., 2011). Therefore, the amount of freezing-induced water migration from the saturated zone to the frozen zone is key to understanding the effects of soil freezing on engineering problems and

ecological processes (Yu et al., 2020). In some previous experimental or numerical studies, freezing-induced water migration from groundwater to the frozen zone was obtained by assuming a fixed water table (Shoop and Bigl, 1997;Hansson and Lundin, 2006;Alkhaier et al., 2012;Chen et al., 2019). It was found that a fixed head lower boundary condition, even if the fixed head is close enough to the mean water table, would overestimate groundwater loss to support evapotranspiration because such a boundary condition implies that loss of groundwater can be replenished instantaneously (Zhu et al., 2010). However, the degree of overestimation of freezing-induced water migration by using a fixed head lower boundary condition remains unknown.

As initially proposed by Hubert (1940) and Tóth (1962), lateral groundwater flow is ubiquitous in regions with undulating topography and water table. Although wintertime water table fluctuations caused by freezing and thawing have been widely recognized (Willis et al., 1964;van der Kamp et al., 2003;Zhang et al., 2019), the simultaneous contribution of lateral groundwater flow and soil freezing-thawing to wintertime water table fluctuations received little attention. Ireson et al. (2013) found the wintertime water table recessions in their two study sites in the Canadian prairies were partly due to lateral groundwater outflow, and Jiang et al. (2017) found the pattern of wintertime water table fall-rise in their study site in the Ordos Plateau, China was also influenced by lateral groundwater inflow. Therefore, the wintertime water table dynamics is a combined product by atmospheric conditions and lateral groundwater inflow/outflow, instead of a prior known fixed boundary condition.

To obtain a clear understanding of wintertime water migration and water budget in regions with shallow groundwater, it is appealing to numerically combine hydrological processes in both saturated and unsaturated zones, including topographically-driven lateral flow in the saturated zone, freezing-induced water migration from the saturated to the unsaturated zone, and the thawing-induced water movement. In the current study, the field site reported in Jiang et al. (2017), with shallow water table and lateral groundwater inflow, is examined as an example. We also use a series of scenarios with different water table depths and lateral groundwater inflow rates to numerically investigate their controls on water table fluctuations and freezing-induced water gain in the frozen zone.

## 2 Methods

### 2.1 Study area and conceptual model

The Ordos Plateau in Northwestern China has a semi-arid and seasonally-frozen climate, and is well-known for the occurrence of topography-driven regional groundwater flow (Hou et al., 2010;Jiang et al., 2018). The main

aquifer is the thick Cretaceous sandstone, which is overlain by a thin layer of sand. Due to the spatially undulating topography and low permeability of the sandstone aquifer, water table undulation is a subdued replica of the topography. In the scale of a catchment, groundwater generally flows laterally from topographic highs to topographic lows, with a downward component of groundwater flow in the recharge area and an upward component in the discharge area. In the discharge area, as a result of the contribution of lateral groundwater flow from topographic highs towards the lake, water table is shallow near the lake. The Wudu Lake catchment (Wang et al., 2015;Zhao et al., 2020) is one of the small catchments located to the southeast of the first-order water divide of the Ordos Plateau. Based on measurements of 53 wells in the catchment (the location of wells can be found in Jiang et al., 2018), water table depth increases from less than 0.5 m in topographic lows to 26.1 m near the divide. In topographic lows with water table ranging from around 1 m to around 2 m, the typical vegetation is *Achnatherum splendens*.

The Otak national meteorological station, which is around 35 km away from the center of the Wudu Lake catchment, is the nearest meteorological station with long-term meteorological observations. From 1955 to 2016 (data available at http://data.cma.cn), the annual mean precipitation is 265.0 mm, including an annual mean snowfall of 16.0 mm, and the annual mean pan evaporation is 1370.0 mm. Note that the amount of snowfall is determined by weighting water equivalent of snowfall, and the accuracy of precipitation and pan evaporation measurements is ±0.1 mm. July has the largest monthly mean temperature equaling 22.5 ℃ while January has the lowest monthly mean temperature equaling -10.4 ℃. The period from late November to late March corresponds to the freezing-thawing cycle. As shown in Jiang et al. (2018), freezing-induced groundwater level decline occurs in the monitoring well DK2 located in topographic lows, but does not occur in the monitoring well DK1 located in topographic highs where water table is deep.

In the Wudu lake catchment, the occurrence of numerous flowing wells in topographic lows indicates an upward hydraulic gradient (Wang et al., 2015), which also induces groundwater inflow into the shallow part of the aquifer. At the monitoring site, lateral groundwater inflow had been found to be an indispensable component of groundwater budget (Jiang et al., 2017). Fig. 1 shows the control of main hydrological and hydrogeological processes on groundwater level during the freezing-thawing cycle. The processes leading to water level decline include water loss induced by freezing and evaporation, and the processes leading to water level rise are lateral groundwater inflow throughout the whole period and infiltration of thawed water during the thawing stage.

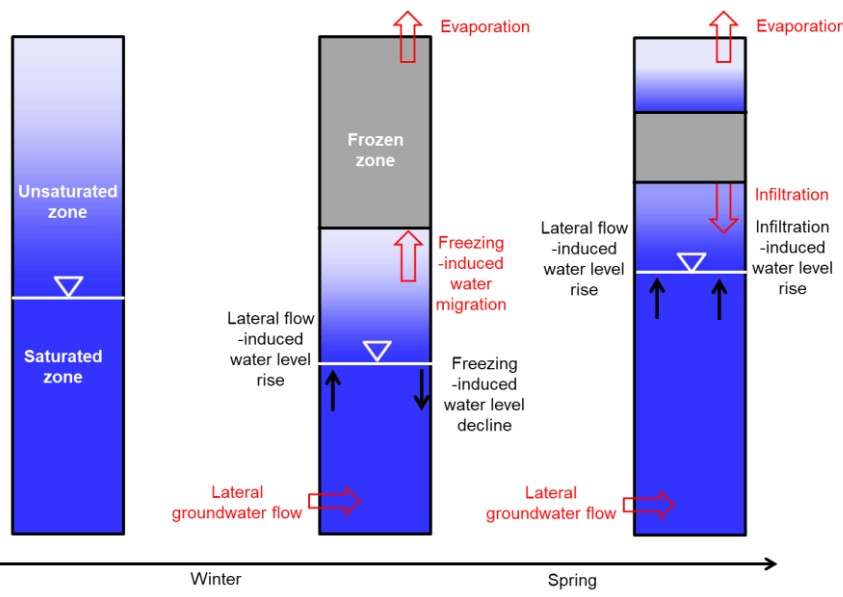

Figure 1 The conceptual model showing main hydrological and hydrogeological processes in semi-arid regions during the freezing-thawing cycle.

**2.2 Field measurement**

To examine how atmospheric conditions control groundwater dynamics in topographic lows, we set up a monitoring profile of soil water contents and soil temperatures adjacent to the groundwater level monitoring well DK2. Soil temperatures and liquid water contents in the Quaternary sands are recorded at 8 depths (at around 10 cm, 20 cm, 30 cm, 50 cm, 70 cm, 90 cm, 110 cm, 150 cm below surface) by using 5TM sensors produced by Decagon Devices. The 5TM sensors, which measure the soil dielectric permittivity to represent liquid soil water content (Zheng et al., 2015), have an accuracy of around ±2% volumetric water content. To ensure the

representativeness of measurement, we performed site-specific calibration by establishing the correlation between liquid water content measured by the 5TM sensors and that measured by the gravimetric method. It has been reported that 5TM sensors are accurate enough to measure the liquid water content even if soil is partially frozen (Yang et al., 2013;Zheng et al., 2017;Xue et al., 2020).

         To obtain the bulk density, 12 soil samples were collected by the cutting-ring method with a maximum

depth of 120 cm and a resolution of 10 cm (Table 1). We also collected soil samples from the 12 depths for measuring soil particle size by using the Mastersizer 2000 instrument (Malvern Instruments, England). The saturated hydraulic conductivity of soil samples from selected layers were measured by the HYPROP (www.metergroup.com/environment/products/hyprop-2/). The soil samples with low clay content above 70 cm and below 100 cm were measured to be around 43.2 m/d, while that with higher clay content at the depth ranging

between 70 and 100 cm is measured to be 1.9 m/d.

In this study, the data collected during 28 NOV 2015 and 1 APR 2016 is used as an example. As shown in Fig. 2, water table depth increases from 115 cm on 28 NOV 2015 to 143 cm on 29 JAN 2016, which corresponds to the stage with an increasing frost depth.

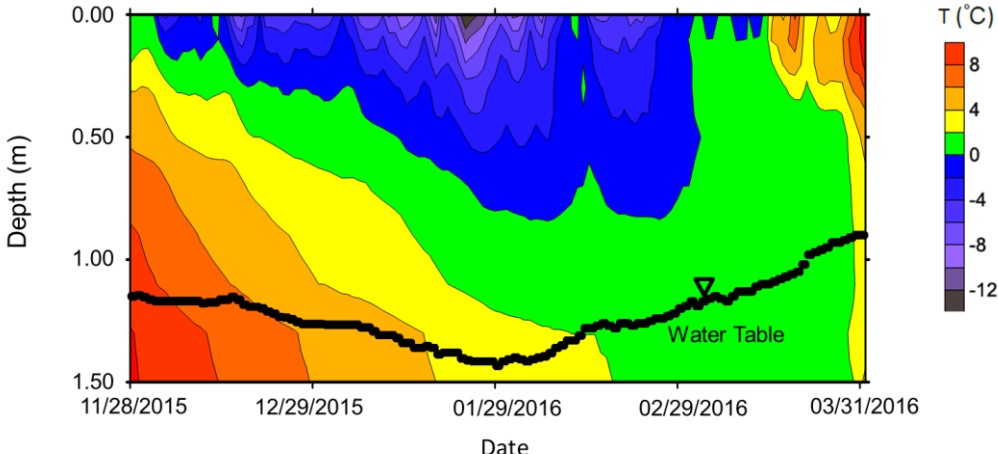

Figure 2. The contour map showing the evolution of soil temperature at the study site during the freezing-thawing cycle. Also shown is the monitored fluctuating groundwater level.

### 2.3. The SHAW model

By assuming that the mechanisms of water transport in partially frozen soils are similar to those in unsaturated soils, Harlan (1973) pioneered modeling studies of heat-fluid transport with freezing-thawing by considering the co-existence of a frozen zone, an unsaturated unfrozen zone and a saturated unfrozen zone. The Simultaneous Heat and Water (SHAW) model (Flerchinger and Saxton, 1989) is one of the most robust Harlan-type models for freezing-induced water migration in the one-dimensional (1D) domain (Hayhoe, 1994;DeGaetano et al., 2001;Kahimba et al., 2009;Li et al., 2010;Zhang et al., 2017b). A 1D soil column is justified to account for water redistribution induced by freezing and thawing because water flow in the unsaturated zone is predominantly vertical (Stephens, 1996;Romano et al., 1998;Van Dam and Feddes, 2000). The contributions of liquid water, ice, and vapor contents have been considered in the water balance equation, and the contributions of heat conduction, phase change, liquid flow, and vapor gas diffusion have been considered in the energy balance equation. By adding a source term representing the flux of groundwater inflow/outflow in the water balance equation, it has the ability to simultaneously account for the contributions of lateral groundwater flow and freezing/thawing in the soil column. Therefore, the SHAW model is suitable to couple the groundwater dynamics with soil water dynamics during the freezing-thawing cycle as observed in our field site.

Considering convective heat transfer by liquid, latent heat transfer by ice and latent heat transfer by vapor, the energy balance equation in the soil matrix is expressed as

$$C_s \frac{\partial T}{\partial t} - \rho_i L_f \frac{\partial \theta_i}{\partial t} = \frac{\partial}{\partial z} \left[ k_T \frac{\partial T}{\partial z} \right] - \rho_l c_l \frac{\partial q_l T}{\partial z} - L_v \left[ \frac{\partial q_v}{\partial z} + \frac{\partial \rho_v}{\partial t} \right] \qquad (1)$$

where $T$ is temperature [$\Theta$], $\theta_i$ is volumetric ice content [-], $z$ is soil depth [L], $t$ is time [T], $\rho_i$, $\rho_l$ and $\rho_v$ are densities of ice, liquid water and vapor [M L$^{-3}$], respectively, $L_f$ and $L_v$ are latent heat of fusion and vaporization [L$^2$ T$^{-2}$], respectively, $q_l$ is liquid water flux [L T$^{-1}$], $q_v$ is water vapor flux [M L$^{-2}$ T$^{-1}$], $c_l$ is specific heat capacity of water [L$^2$ T$^{-2}$ $\Theta^{-1}$], $C_s$ is volumetric heat capacity of soil [M L$^{-2}$ T$^{-2}$ $\Theta^{-1}$], and $k_T$ is thermal conductivity of soil [M L$^2$ T$^{-3}$ $\Theta^{-1}$]. $C_s$ of soil is determined by the sum of the volumetric heat capacities of the soil constituents:

$$C_s = \sum \rho_j c_j \theta_j \qquad (2)$$

and $k_T$ of soil is calculated by the following equation proposed by de Vries (1963):

$$k_T = \frac{\sum m_j k_j \theta_j}{\sum m_j \theta_j} \qquad (3)$$

where $\rho_j$, $c_j$, $\theta_j$, $k_j$ and $m_j$ are the density, specific heat capacity, volumetric fraction, thermal conductivity and weighting factor of the $j^{\text{th}}$ soil constituent (sand, silt, clay, water, ice and air).

By considering ice content change and vapor flux, the water balance equation is written as

$$\frac{\partial \theta_l}{\partial t} + \frac{\rho_i}{\rho_l} \frac{\partial \theta_i}{\partial t} = \frac{\partial}{\partial z} \left[ K \left( \frac{\partial \psi}{\partial z} + 1 \right) \right] + \frac{1}{\rho_l} \frac{\partial q_v}{\partial z} + U \qquad (4)$$

where $\theta_l$ is liquid water content [-], $K$ is hydraulic conductivity [L T$^{-1}$], $\psi$ is soil matric potential [L], and $U$ is a source/sink term for water flux [T$^{-1}$]. The lateral inflow rate per unit volume, which is one form of source, is calculated by using the Darcy's law based on a constant horizontal hydraulic gradient at selected nodes within the saturated zone. Soil water characteristic curves, which describe the relationship between soil matric potential and liquid water content, are defined for both unfrozen and frozen soils. Here, the van Genuchten equation (van Genuchten, 1980) is selected, which is written as

$$\frac{\theta_l - \theta_r}{\theta_s - \theta_r} = \left( 1 + |\alpha \psi|^n \right)^{-m} \qquad (5)$$

where $\theta_s$ and $\theta_r$ are saturated and residual water content [-], $\alpha$ [L$^{-1}$], $n$ [-] and $m$ [-] are empirical parameters. $\alpha$ equals the inverse of the air-entry value, $n$ is the pore-size distribution index and $m=1-1/n$. In the frozen zone with co-existence of liquid water and ice, the matric potential, which is strongly dependent on temperature, can be obtained by the following equation (Fuchs et al., 1978):

$$\psi = \frac{L_f}{g} \frac{T}{T + 273.15} \qquad (6)$$

where $g$ is the acceleration of gravity [L T$^{-2}$]. Eq. (6) indicates that as the negative temperature increases, the soil suction also increases (matric potential becomes more negative).

In the unfrozen zone, the hydraulic conductivity is computed by

$$K = K_s S_e^l \left[ 1 - \left( 1 - S_e^{1/m} \right)^m \right]^2 \tag{7}$$

where $K_s$ is the saturated hydraulic conductivity [L T$^{-1}$], $S_e$ [-] is the effective saturation calculated by $(\theta_l - \theta_r)/(\theta_s - \theta_r)$, and $l$ [-] is the pore-connectivity parameter which is assumed to be 0.5 (Mualem, 1976). In the frozen zone, occupation of ice in the pores would decrease the available porosity and thus hydraulic conductivity. When the available porosity is lower than 0.13, $K$ is assumed to be 0; when the available porosity is above 0.13, $K$ computed from Eq. (7) is reduced linearly with ice content assuming zero conductivity at an available porosity of 0.13 (Flerchinger, 2000).

The freezing-thawing cycle is usually accompanied by occasional or frequent snowfall events. In the SHAW model, precipitation is assumed to be snowfall if the air temperature is below 0℃. When snow falls on bare soil with a surface temperature of below 0 ℃, a snow cover layer is formed. The energy balance for the snow cover is written as follows (Flerchinger, 2000):

$$\rho_{sp} c_i \frac{\partial T}{\partial t} + \rho_l L_f \frac{\partial w_{sp}}{\partial t} = \frac{\partial}{\partial z} \left[ k_{sp} \frac{\partial T}{\partial z} \right] + \frac{\partial R_n}{\partial z} - L_s \left[ \frac{\partial q_v}{\partial z} + \frac{\partial \rho_v}{\partial t} \right] \tag{8}$$

where $\rho_{sp}$ is the density of snow [M L$^{-3}$], $w_{sp}$ is the liquid water content in the snow [-], $c_i$ is the heat capacity of ice [L$^2$ T$^{-2}$ Θ$^{-1}$], $L_s$ is the latent heat of sublimation [L$^2$ T$^{-2}$], $k_{sp}$ is the thermal conductivity of snow [M L$^2$ T$^{-3}$ Θ$^{-1}$], $R_n$ is the net downward radiation flux within the snow [M L$^2$ T$^{-4}$]. If the air temperature is increased to above 0℃, snow got melted. Snowmelt in excess of the calculated interception could infiltrate into the subsurface medium when the maximum depth of ponding is set to be a positive value.

## 2.4. Model inputs

Based on the conceptual model shown in Figure 1, we use SHAW to couple observed soil water and groundwater dynamics and wintertime atmospheric conditions during the freezing-thawing cycle from 28 NOV 2015 to 1 APR 2016. The time step of numerical simulation is one hour. Because the deepest water table depth is 143 cm and soil temperature at 150 cm below surface is available, the length of the 1D soil column is set to be 150 cm. The model domain is uniformly divided into 30 layers, which results in 31 nodes. The stable variables of the model include soil temperature and soil water content. The top node has an atmospheric boundary condition, while the bottom node has a zero-flux and specified-temperature boundary condition. The occurrence of lateral groundwater flow is realized by specifying a hydraulic gradient at a node within the saturated zone (the 30[th] node in the current study). In this way, the fluctuating groundwater level is an outcome of wintertime atmospheric conditions and lateral groundwater inflow, while the fluctuating soil water content is controlled by both

wintertime atmospheric conditions and groundwater dynamics.

The inputs for the atmospheric boundary conditions include maximum air temperature, minimum air temperature, wind speed, precipitation, solar radiation and dew-point temperature. The first four parameters are available from the observations at the Otak meteorological stations, the parameter of solar radiation is calculated by the Angström-Prescott equation (Yorukoglu and Celik, 2006) based on the sunshine duration, and the

dew-point temperature is calculated by the Hyland-Wexter equation (Wexler et al., 1983) based on relative humidity, maximum air temperature and minimum air temperature. Fig. 3 shows the maximum and minimum air temperatures, dew-point temperature and precipitation. The inputs for the lower boundary condition at the bottom include a zero hydraulic conductivity to obtain a no-flux boundary condition, and the daily averaged soil temperature measured at 150 cm below surface for the specified-temperature boundary condition. The initial

conditions of soil water content and soil temperature for spin-up are determined based on measured values on 29 OCT 2015, and the model spin-up is run for 30 days before the start of freezing stage (28 NOV 2015). .

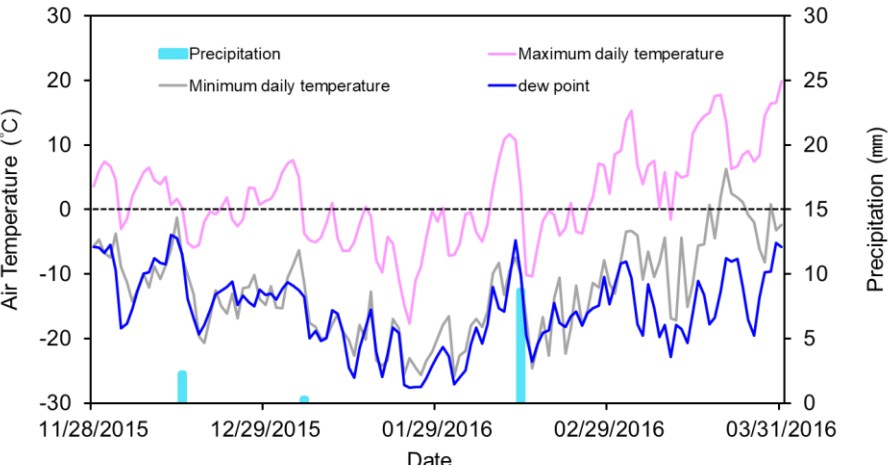

Figure 3 The maximum daily air temperature, the minimum daily air temperature, dew-point temperature and precipitation during the freezing-thawing cycle from 28 NOV 2015 to 1 APR 2016.

Soil texture determines the thermal parameters and the initial hydraulic parameters. As shown in Table 1, the percentage of clay ranges between 5% and 8% in the layer between 70 and 100 cm, and between 1% and 4% in other depths. Therefore, the model domain is divided into three layers. Based on average contents of clay, silt and sand in each layer, the thermal parameters are calculated by using equations (2) and (3). The hydraulic

parameters ($\theta_r$, $\theta_s$, $\alpha$, $n$), which are initially estimated by the Rosetta pedotransfer function (Schaap and Leij, 1998;Zhang et al., 2017a), are further calibrated to fit the measured liquid soil water content (Table 1). Based on the possible range of lateral inflow rate varying between 0.96 and 1.16 mm/d (Jiang et al., 2017), the lateral inflow rate is calibrated to be 1.03 mm/d, which leads to the lowest RMSE of water table depth.

Table 1 The measured soil texture and bulk density, and calibrated hydraulic parameters.

| Depth (cm) | Clay (%) | Silt (%) | Sand (%) | Bulk density (g/cm$^3$) | $\theta_r$ (cm$^3$/cm$^3$) | $\theta_s$ (cm$^3$/cm$^3$) | $\alpha$ (m$^{-1}$) | $n$ (-) | $K_s$ (m/d) |
|---|---|---|---|---|---|---|---|---|---|
| 10 | 1.5 | 9.4 | 89.1 | 1.638 | | | | | |
| 20 | 1.6 | 9.5 | 88.9 | 1.673 | | | | | |
| 30 | 1.5 | 9.5 | 89.0 | 1.628 | 0.012 | 0.408 | 4.1 | 2.06 | 3.84 |
| 40 | 2.0 | 9.5 | 88.5 | 1.672 | | | | | |
| 50 | 2.5 | 9.1 | 88.3 | 1.655 | | | | | |
| 60 | 2.7 | 9.7 | 87.6 | 1.613 | | | | | |
| 70 | 5.5 | 13.5 | 81.0 | 1.562 | | | | | |
| 80 | 6.4 | 11.2 | 82.4 | 1.518 | 0.076 | 0.381 | 3.6 | 1.86 | 0.41 |
| 90 | 6.1 | 9.2 | 84.7 | 1.549 | | | | | |
| 100 | 7.7 | 10.4 | 81.9 | 1.598 | | | | | |
| 110 | 3.0 | 9.4 | 87.6 | 1.652 | | | | | |
| 120 | 1.7 | 9.7 | 88.6 | 1.733 | | | | | |
| 130 | - | - | - | - | 0.032 | 0.403 | 6.4 | 2.26 | 3.41 |
| 140 | - | - | - | - | | | | | |
| 150 | - | - | - | - | | | | | |

To examine the role of lateral groundwater inflow and initial water table depth in freezing-induced groundwater migration, we build different scenarios with three different initial water table depths (120 cm, 170 cm, and 220 cm) and three different lateral flow rates (0, 0.51 mm/d, and 1.03 mm/d) for sensitivity analysis. We also build one scenario with an initial water table depth of 250 cm but without lateral groundwater inflow, and three scenarios with water table depths fixed at 120 cm, 170 cm, and 220 cm. The specific settings of the 13 scenarios are shown in Table 2. To ensure that water table can be explicitly represented in the models, the length of model domain is extended to 250 cm. To exclude the influence of heterogeneity, the whole soil column is assumed to be filled with sand of the same soil texture as that of 0 cm to 70 cm in the field site. Because there is no temperature measurement at 250 cm below surface, the temperature at the lower boundary is estimated by the force-restore approach embedded in the SHAW model (Hirota, 2002), which can be written as:

$$\left(1+\frac{2z}{d_d}\right)\frac{\partial T}{\partial t}=\frac{2}{C_S d_d}G-\omega\left(T-T_{AVG}\right) \tag{9}$$

where $z$ is the depth [L] below the surface, $\omega$ is the frequency [T$^{-1}$] of fluctuation period, $d_d$ is damping depth [L] corresponding to $\omega$, which is expressed as $d_d=\left(\dfrac{2k_T}{C_S\omega}\right)^{1/2}$, and $T_{AVG}$ is the average annual air temperature [Θ].

Table 2 The different scenarios of sensitivity analysis with different initial water table depths and different conditions of groundwater supply.

| Scenarios | Initial Water Table Depth (cm) | Groundwater supply |
|---|---|---|
| Scenario A1 | 120 | LFR*=0 |
| Scenario A2 | 120 | LFR=0.51 mm/d |
| Scenario A3 | 120 | LFR=1.03 mm/d |
| Scenario A4 | 120 | Determined by the fixed water table |
| Scenario B1 | 170 | LFR=0 |
| Scenario B2 | 170 | LFR=0.51 mm/d |
| Scenario B3 | 170 | LFR=1.03 mm/d |
| Scenario B4 | 170 | Determined by the fixed water table |
| Scenario C1 | 220 | LFR=0 |
| Scenario C2 | 220 | LFR=0.51 mm/d |
| Scenario C3 | 220 | LFR=1.03 mm/d |
| Scenario C4 | 220 | Determined by the fixed water table |
| Scenario D | 250 | LFR=0 |

*LFR represents lateral flow rate.

## 3 Results and discussion

### 3.1 Field data showing freezing-induced water migration and water level decline

The 0 ℃ isothermal contour of soil temperature was shown in Fig. 2, which well characterizes the gradual deepening of frost depth during the freezing stage from 28 NOV 2015 to 29 FEB 2016. Nevertheless, soil temperature alone is not able to demonstrate freezing-induced water migration. Fig. 4 shows the evolutions of liquid water content and soil temperature at 6 depths (10 cm, 20 cm, 30 cm, 50 cm, 70 cm, and 90 cm). The dates of the start of freezing, i.e., when soil temperature drops to 0 ℃, at 10 cm, 20 cm, 30 cm, 50 cm, and 70 cm are 02 DEC, 15 DEC, 16 DEC, 9 JAN and 23 JAN, respectively (Fig. 4). At 10 cm, 20 cm and 30 cm below surface, there is a sharp decrease in liquid water content when the soil temperature drops to 0 ℃, while at 50 cm and 70 cm below surface, due to freezing-induced water migration toward the freezing front, there is a smooth decrease in liquid water content before reaching 0 ℃. At 90 cm, probably because the freezing front is close enough to the sensor, although the temperature at the sensor does not reach 0 ℃, there is a significant decrease in liquid water content on 29 FEB due to freezing-induced water redistribution. Therefore, the relationship between temperature and liquid water content recorded by 5TM sensors, which are based on the frequency domain reflectometer (FDR) method, well reflects the effect of freezing on liquid water content.

Due to the rising air temperature in spring, soil temperature at 10 cm increased from below 0 ℃ to above

0 ℃ on 1 MAR, which is considered to be the start of the thawing stage in the current study. Because 1 APR has the highest water level as a result of the downward infiltration of thawed water, this date is considered to be the

end of the thawing period. For the sensors at 20 cm and 70 cm, the start date of thawing stage is 2 MAR, while for the sensors at 30 cm and 50 cm, the start dates of thawing stage are 3 MAR, and 5 MAR, implying the occurrence of bi-directional thawing. Different from the quick response of soil temperature to freezing, the response of soil temperature to thawing is a slow process. For the 3 sensors at 10 cm, 20 cm and 30 cm, the duration from start to end of thawing is 13 days, while for the 2 sensors at 50 cm and 70 cm, the durations from

start to end of thawing are 18 days and 26 days, respectively. The liquid water content increases sharply at the end of thawing and is much higher than that before freezing as a result of the contribution of freezing-induced water gain.

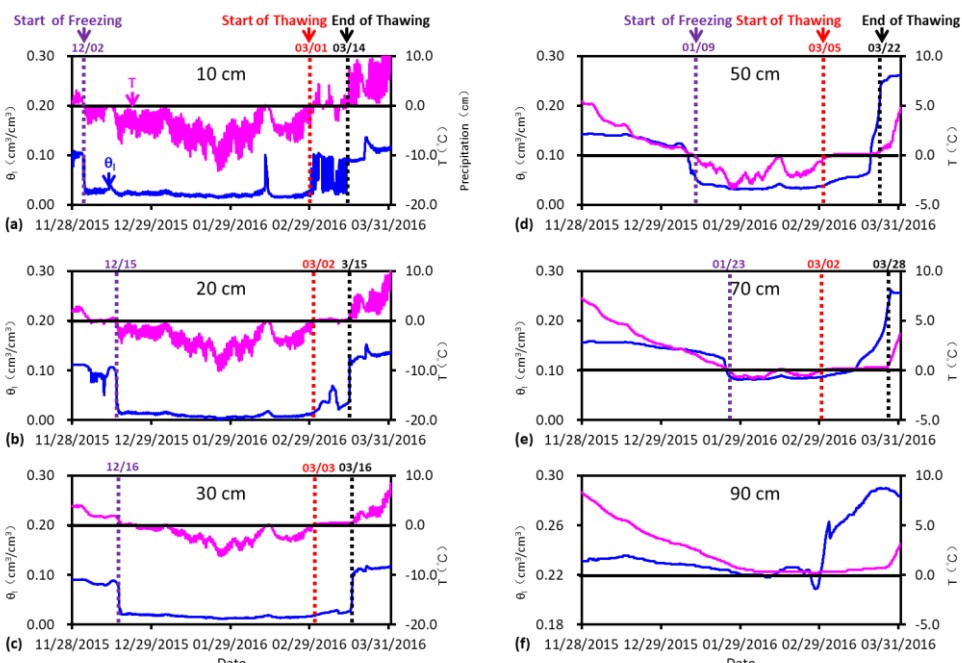

Figure 4 The observed hourly liquid water content and soil temperatures during the freezing-thawing cycle from
28 NOV 2015 to 1 APR 2016.

As shown in Fig. 1, water table depths on 28 NOV and 1 APR are 115 cm and 90 cm, respectively. Because snowfall in winter is only 11.7 mm, the much higher water level at the end of thawing stage is mainly due to the occurrence of lateral groundwater inflow as reported in Jiang et al. (2017). The lowest water level of 143 cm below surface occurs on 29 JAN. The trend of rising water level from 29 JAN to 29 FEB, when frozen soil has

not been thawed yet, is attributed to lateral groundwater inflow. The occurrence of lateral groundwater inflow also alleviated the magnitude of freezing-induced groundwater level decline from 28 NOV to 29 JAN, i.e., freezing-induced groundwater level decline should be greater than 28 cm if there is no lateral inflow. To quantify

the water budget during the freezing-thawing cycle, we applied the SHAW model to determine the fluxes of freezing-induced migration, lateral groundwater inflow, infiltration and soil evaporation.

**3.2 Numerical results of water redistribution during the freezing-thawing cycle**

Based on the parameters listed in Table 1, and with a lateral groundwater inflow rate of 1.03 mm/d, there are good agreements between simulated and measured liquid water content in the frozen zone (Fig. 5a), between simulated and measured soil temperature (Fig. 5b), as well as between simulated and measured groundwater level (Fig. 6). Fig. 5a also shows the simulated total water content, which coincides with liquid water content before freezing, but increases rapidly after freezing and finally maintains at a stable value. The rapid increase in total water content is due to freezing-induced water migration from the underlying soil.

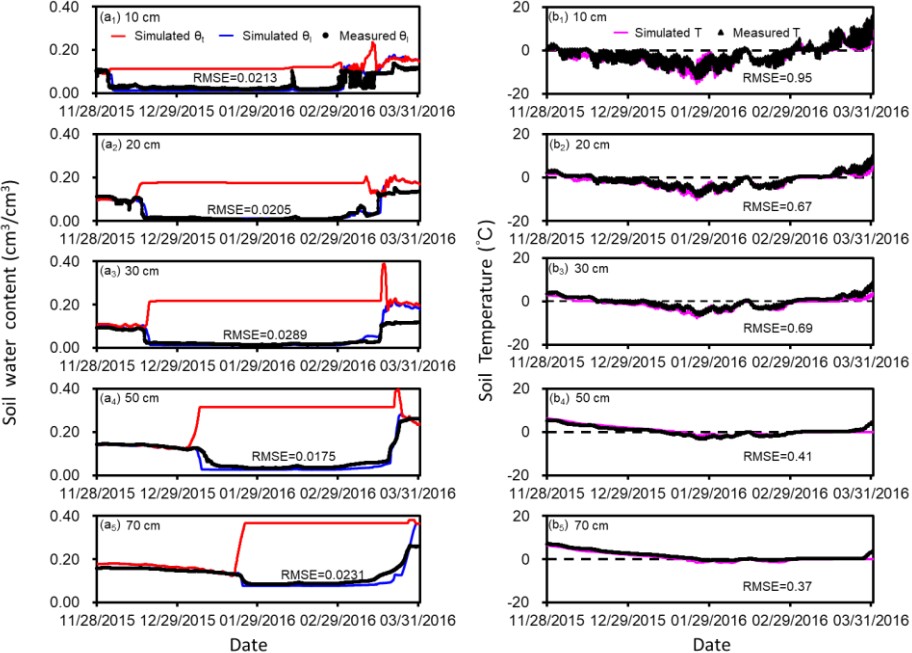

Figure 5 The observed and simulated liquid water content and simulated total water content ($\theta_t$) ($a_1$-$a_5$) and soil temperature ($b_1$-$b_5$) at the 5 different depths within the frozen zone during the freezing-thawing cycle. The RMSEs of liquid water content and soil temperature are also listed.

Fig. 6a shows the distribution of simulated soil temperature and frost depth. There is a good match between simulated and measured frost depth, the latter of which is determined by the dates of start of freezing and end of thawing as shown in Figure 4. In the frozen zone, as the frost depth increases, ice content increases from the shallow to the deep, but liquid water content is generally maintained around the residual water content (Fig. 6b,c). In the node above the frost depth, the ice content is lower than the overlying node, and the liquid water content is higher than the residual water content. Note that when the frost depth reaches the middle layer with finer soils, there is a thicker zone with lower ice content and higher liquid water content, which is related to

the higher residual water content of the middle layer. In the whole frozen zone, the trend of depth-increasing total

water content toward the water table (Fig. 6d) indicates a trend of depth-increasing water gain controlled by the

distance between the freezing front and the water table.

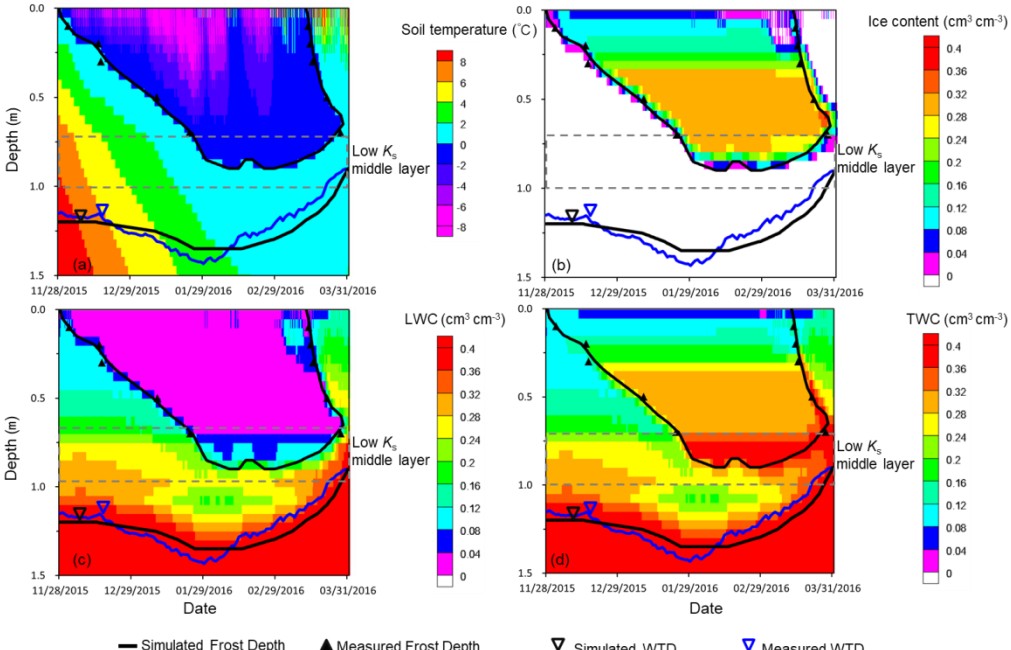

Figure 6 The evolutions of simulated soil temperature (a), ice content (b), liquid water content (LWC) (c) and total water content (TWC) (d) at the field site. Also shown in the figures are evolutions of frost depth and groundwater level.

**3.3 Water budgets during the freezing-thawing cycle: implications for appropriate boundary conditions**

During the freezing-thawing cycle, infiltration of snowmelt and lateral groundwater inflow are two

processes of water gain in the soil column, while soil evaporation is the only form of water loss leaving the top of

the soil column. At our field site, the cumulative infiltration of snowmelt is simulated to be 3.2 mm, while the

cumulative evaporation is simulated to be 32.0 mm, both of which are restricted by the frozen soil. Based on the

temporal evolution of evaporation rate (Fig. 7), we identify three stages of soil water evaporation. Stage Ⅰ has a

trend of decreasing evaporation rate, stage Ⅱ has a limited evaporation rate, and stage Ⅲ has a trend of

increasing evaporation rate. Note that in stage Ⅱ the atmospheric conditions associated with the snowfall event

on 13 FEB result in slightly higher evaporation rate. The mean evaporation rates in the three stages are 0.36, 0.18

and 0.59 mm/d, respectively. Similar three stages of temporal evolution of soil evaporation during the

freezing-thawing cycle have been reported in several field studies (Kaneko et al., 2006;Wu et al., 2016;Xue et al.,

2020). The high evaporation in stage Ⅲ, which is a direct result of accumulation of liquid water during the

thawing stage, well explains the occurrence of soil salinization in spring in many regions of the world (Liu et al.,

2009;Lopez et al., 2010;Bechtold et al., 2011) .

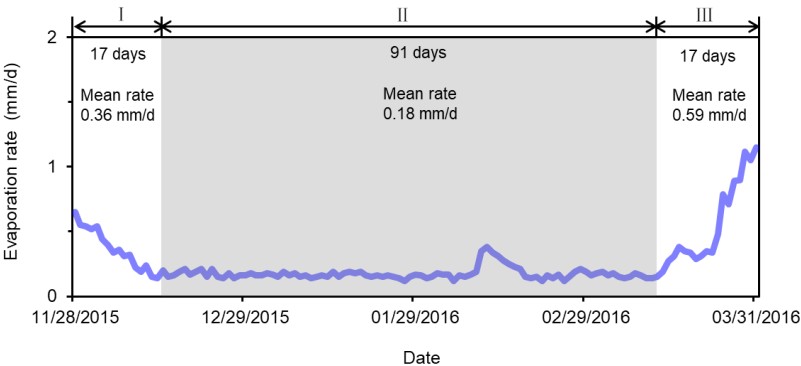

Figure 7 The simulated daily soil evaporation rate at the field site . Also shown are the mean evaporation rates for the three stages with different trends of daily soil evaporation rates.

If the freezing-thawing cycle is divided into the freezing stage and the thawing stage, the balance of water in each stage can be written as

$$S_{s0} + S_{u0} + q \cdot \Delta t_F + I_F = S_{1s} + S_{1u} + ET_F \qquad (10a)$$

$$S_{s1} + S_{u1} + q \cdot \Delta t_T + I_T = S_{s2} + S_{u2} + ET_T \cdot \Delta t_T \qquad (10b)$$

where $S_{s0}$ [L], $S_{s1}$ [L] and $S_{s2}$ [L] are cumulative water storages in the saturated zone at the beginning of the freezing stage, at the end of the freezing stage (the beginning of the thawing stage) and at the end of the thawing

stage, respectively; $S_{u0}$ [L], $S_{u1}$ [L] and $S_{u2}$ [L] are cumulative water storages in the unsaturated zone at the beginning of the freezing stage, at the end of the freezing stage (the beginning of the thawing stage) and at the end of the thawing stage, respectively; $q$ [L/T] is the mean rate of lateral groundwater inflow during the freezing-thawing cycle; $\Delta t_F$ [T] and $\Delta t_T$ [T] are the durations of the freezing and thawing stages, respectively; $I_F$ [L] and $I_T$ [L] are the cumulative snowmelt infiltration during the freezing and thawing stages, respectively; $ET_F$

[L/T] and $ET_T$ [L/T] are the mean soil evaporation rates during the freezing and the thawing stages, respectively. The values of each term in Eq. (10) are listed in Table 3.

Table 3 Water budgets during the freezing-thawing cycle

| Stage | Duration | Initial $S_s$ (mm) | Initial $S_u$ (mm) | $q \cdot \Delta t$ (mm) | $I$ (mm) | $ET \cdot \Delta t$ (mm) | Final $S_s$ (mm) | Final $S_u$ (mm) | $\Delta S_s$ (mm) |
|---|---|---|---|---|---|---|---|---|---|
| Freezing | 28 NOV 2015 - 28 FEB 2016 (93 days) | 120.9 | 246.4 | 95.8 | 3.2 | 19.5 | 80.6 | 366.2 | -136.1 |
| Thawing | 29 FEB 2016 - 1 APR 2016 (32 days) | 80.6 | 366.2 | 33.0 | 0 | 12.5 | 239.6 | 227.7 | 126.0 |

Apparently, the change in groundwater storage, $\Delta S_s$, during the freezing stage can be calculated by $S_{s1} - S_{s0}$, which equals -40.3 mm and corresponds to a water table decline of 10 cm. In fact, such a water table decline has been alleviated by the lateral groundwater inflow. The change in groundwater storage, $\Delta S_s$, should be calculated by $S_{s1} - S_{s0} - q \cdot \Delta t_F$, which equals -136.1 mm, i.e., freezing-induced groundwater loss should be 136.1 mm. In a groundwater flow model without considering the unsaturated zone, this process can be characterized by a mean upward groundwater flux of 1.46 mm/d with a duration of 93 days at the upper boundary. Similarly, the change in groundwater storage, $\Delta S_s$, in the thawing stage should be calculated by $S_{s2} - S_{s1} - q \cdot \Delta t_T$, which equals 126.0 mm. In a groundwater flow model, this process can be characterized by a mean downward groundwater flux of 3.94 mm/d at the upper boundary with a duration of 32 days.

As demonstrated in the pioneering studies of regional groundwater flow (Hubbert, 1940; Toth, 1962), shallow groundwater in the discharge area is dominated by upward flow due to strong evapotranspiration. The above calculation confirms that during 3/4 of the freezing-thawing period, shallow groundwater is also dominated by upward flow. As demonstrated in Jiang et al. (2017), at our study site in the unfrozen period, the stage from late SEP 2014 to early OCT 2014 (with a duration of 18 days) is the only stage with net downward groundwater flux equaling 2.99 mm/d. The current study indicates that the thawing stage has an even larger downward groundwater flux and an even longer duration. Therefore, it is interesting to examine the pattern of two-dimensional or three-dimensional groundwater flow induced by the freezing-thawing cycle by using a fully coupled soil-groundwater model in the future.

## 3.4 The effect of water table depth and lateral groundwater inflow on freezing-induced water redistribution

In Subsect.3.1, we inferred that due to the occurrence of lateral groundwater inflow, freezing-induced groundwater level decline is alleviated, and there is a trend of groundwater level rise before the end of the freezing stage. Figure 8a shows the evolutions of total soil water content, frost depth and groundwater level under four conditions of groundwater inflow when the initial water table depth ($WTD_0$) equals 120 cm. When there is no lateral groundwater inflow (Fig. 8a1), the lowest water level occurs at the end of the freezing period, corresponding to a water table decline of 45 cm. A comparison of water table fluctuations under three different rates of lateral groundwater inflow (Fig. 8a1-a3) confirmed that the water level rise before the start of the thawing stage is caused by lateral groundwater inflow. Therefore, in the field, whether the timing of lowest water table corresponds to the end of the freezing stage can be used to infer the existence of lateral groundwater

inflow/outflow. The higher water level caused by lateral groundwater inflow would be beneficial for freezing-induced water migration towards the freezing front. As lateral groundwater inflow rate changes from 0 to 0.51 to 1.03 mm/d, the increase in water storage in the frozen zone changes from 108.3 mm to 112.4 mm to 125.6 mm, while the increase in total water content in the frozen zone changes from 0.114 $cm^3/cm^3$ to 0.118 $cm^3/cm^3$ to 0.132 $cm^3/cm^3$ (Table 4). As shown in Fig. 8a3, the total water content above the freezing front could be as high as 0.286 $cm^3/cm^3$. We infer that the co-occurrence of shallow water table and lateral groundwater inflow would enhance the possibility of frost heave.

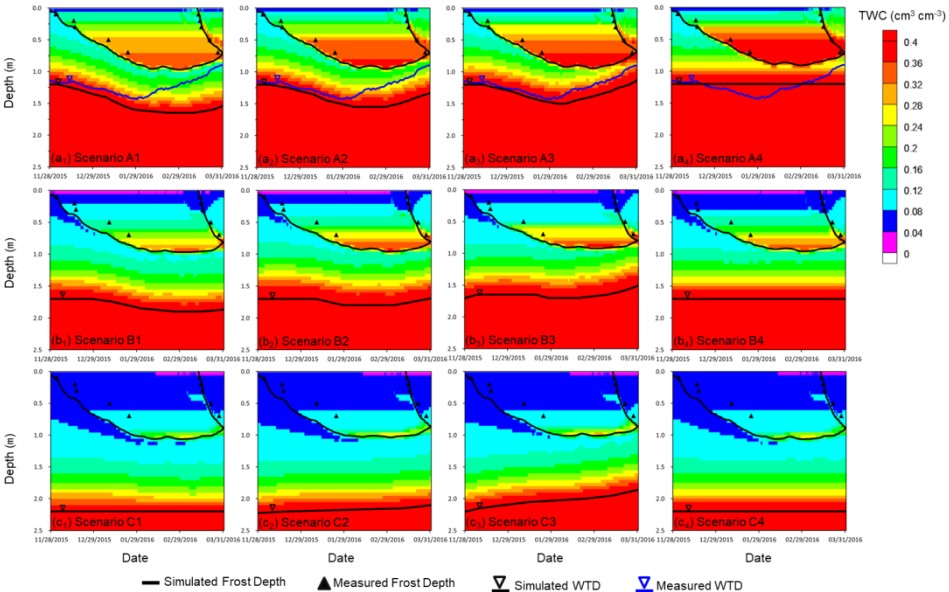

Figure 8 The evolutions of simulated total water content, frost depth and water table depth under three different initial water table depths (120 cm, 170 cm and 220 cm) and four different groundwater supply conditions.

Table 4 A comparison of water table decline (ΔWTD), frost depth (FD), and water gain in the frozen zone under different scenarios.

| Scenarios | $WTD_0$ (cm) | $\Delta WTD$ (cm) | FD (cm) | Initial $S_{FZ}$ (mm) | Final $S_{FZ}$ (mm) | Mean TWC | $\Delta S_{FZ}$ (mm) | $\Delta TWC$ |
|---|---|---|---|---|---|---|---|---|
| Scenario A1 | 120 | 45 | 95 | 146.1 | 254.4 | 0.268 | 108.3 | 0.114 |
| Scenario A2 | 120 | 35 | 95 | 146.1 | 258.5 | 0.272 | 112.4 | 0.118 |
| Scenario A3 | 120 | 30 | 95 | 146.1 | 271.7 | 0.286 | 125.6 | 0.132 |
| Scenario A4 | 120 | - | 90 | 130.6 | 274.4 | 0.305 | 143.8 | 0.160 |
| Scenario B1 | 170 | 20 | 95 | 84.7 | 156.9 | 0.165 | 72.2 | 0.076 |
| Scenario B2 | 170 | 10 | 95 | 84.7 | 161.9 | 0.170 | 77.2 | 0.081 |
| Scenario B3 | 170 | -5 | 95 | 84.7 | 177.6 | 0.187 | 92.9 | 0.098 |
| Scenario B4 | 170 | - | 95 | 84.7 | 163.1 | 0.172 | 78.4 | 0.083 |
| Scenario C1 | 220 | 0 | 105 | 79.4 | 102.0 | 0.097 | 22.6 | 0.022 |
| Scenario C2 | 220 | -5 | 105 | 79.4 | 102.8 | 0.098 | 23.4 | 0.022 |
| Scenario C3 | 220 | -20 | 105 | 79.4 | 105.4 | 0.100 | 26.0 | 0.025 |
| Scenario C4 | 220 | - | 105 | 79.4 | 102.2 | 0.097 | 22.8 | 0.022 |
| Scenario D | 250 | 0 | 115 | 81.6 | 93.2 | 0.081 | 11.6 | 0.010 |

Mean TWC is calculated by total water storage in the frozen zone ($S_{FZ}$) divided by frost depth while ΔTWC, is calculated by increased water storage in the frozen zone ($\Delta S_{FZ}$) divided by frost depth.

It is also interesting to examine the control of initial water table depth on freezing-induced groundwater migration. Fig. 8b and 8c show the evolutions of total water content, frost depth and groundwater level when $WTD_0$ equals 170 cm and 220 cm. If there is no lateral groundwater inflow, freezing-induced groundwater level decline is 20 cm when $WTD_0$ equals 170 cm, and is 0 when $WTD_0$ equals 220 cm and 250 cm. When $WTD_0$ equals 220 cm and 250 cm, the zero water table decline implies that groundwater is not directly involved in freezing-induced water redistribution, but there is still freezing-induced water migration from the unsaturated zone. When $WTD_0$ equals 250 cm, we find the lower soil water content below the freezing front led to an even smaller freezing-induced water migration, with the increased total water content as low as 0.010 cm$^3$/cm$^3$ (Table 4). We also note that as $WTD_0$ increases, the smaller soil water content in the shallow part of the soil column also results in larger frost depths. When the lateral groundwater inflow is fixed at other rates, the increased water storage in the frozen zone also decreases with initial water table depth (Table 4).

When the water table depth is fixed at 120 cm, 170 cm, and 220 cm, the increase in total water content in the frozen zone is found to be 0.160 cm$^3$/cm$^3$, 0.083 cm$^3$/cm$^3$ and 0.022 cm$^3$/cm$^3$, respectively. Compared with the cases without lateral groundwater inflow but with a dynamic water table (Scenarios A1, B1 and C1), a fixed

water table leads to higher water gain in the frozen zone. This indicates that a fixed head lower boundary condition would overestimate freezing-induced water migration as well as the risk of frost heave because such a lower boundary condition implies instantaneously replenishment of groundwater. This comparison also shows the necessity of properly characterizing water table fluctuations induced by freezing and groundwater inflow/outflow.

## 4 Conclusions

Based on field observations of soil temperature, liquid soil water content and groundwater level at a site in the discharge area with shallow water table, it was found that the temporal variations of wintertime liquid water content and groundwater level are not only controlled by freezing and thawing, but also by lateral groundwater inflow. Therefore, we constructed a model to examine the responses of soil water and groundwater to wintertime climatic conditions and lateral groundwater inflow. The observed fluctuating soil water contents and groundwater level are well reproduced by the calibrated model, which increases our understanding of water balance as well as the influencing factors of freezing-induced groundwater migration.

Based on the groundwater budget during the freeing-thawing cycle, the mean flux of freezing-induced groundwater loss in the freezing stage with a duration of three months is 1.46 mm/d, and the mean flux of thawing-induced groundwater gain in the thawing stage with a duration of one month is 3.94 mm/d. As found out by Jiang et al. (2017), due to the semi-arid climate, water flux during the majority of the unfrozen period is controlled by evaporation, and the only stage with net recharge in the unfrozen period is from late September to early October. Therefore, combined with the fluxes obtained by Jiang et al. (2017) for the unfrozen period, the fluxes during the freezing-thawing cycle obtained in the current study can be useful for future studies on transient groundwater flow models to characterize the control of climatic conditions on deeper groundwater flow in semi-arid and seasonally frozen regions.

Based on a series of modeling with different initial water table depths, we found that when water table depth is maintained at 220 cm below surface, although part of water in the unsaturated zone can be migrated to the freezing front, groundwater in the saturated zone is not directly involved in freezing-induced water migration. However, when initial water table depths are shallower than 220 cm, freezing-induced groundwater migration could be significant, and the rise in groundwater level due to lateral groundwater inflow could increase the amount of groundwater migrated to the freezing front. Fan et al. (2013) reported that regions with water table depth shallower than 2 m account for around 31% of the global land area while Zhang et al. (2003) reported that seasonally frozen soils underlie approximately 24% of the Northern Hemisphere exposed land surface. Moreover,

winter-time water table recession has been reported in Canada (van der Camp et al., 2003;Ireson et al., 2013), China (Wu et al., 2016;Zhang et al., 2019), Russia (Vinnikov et al., 1996), Sweden (Stahli et al., 1999) and the USA (Drescher et al., 1955;Schneider, 1961;Daniel and Staricka, 2000), implying that the involvement of groundwater in freezing-induced water redistribution is a common physical process. Although the threshold water

table depth in other study areas with different climate conditions remains unknown, our study well demonstrated the necessity of properly characterizing freezing-induced water table fluctuations to quantify freezing-induced groundwater migration and its effect on engineering problems and ecological processes in cold regions.

Code/Data availability

The numerical models in the current study were built by using the SHAW model (vision 3.0), which is available at https://www.ars.usda.gov/pacific-west-area/boise-id/watershed-management-research/docs/shaw-model/.

The meteorological data of the Otak station is available from China Meteorological Data Service Center (http://data.cma.cn).

The observed soil temperature, liquid water content and water table depth from 28 NOV 2015 to 1 APR 2016 at the field site are available in the supplementary document.

**Author contribution**

XJ designed the field site and HX was involved in data collection. All authors were involved in data interpretation. XJ and LW developed the initial idea of the current study, and HX established the models with inputs from all coauthors. HX and XJ wrote the manuscript with contributions from YZ.

**Competing interests**

The authors declare that they have no conflict of interest.

**Acknowledgements**

This study is supported by the National Natural Science Foundation of China (41772242), the Fundamental Research Funds for the Central Universities of China and the 111 Project (B20010). The authors thank two anonymous reviewers whose comments led to significant improvement of the study.

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
