# Peer review of "Interaction of soil water and groundwater during the freezing-thawing cycle: field observations and numerical modeling"

_Hydrology and Earth System Sciences, 2020_

## Author Comment (AC1)

**Response to Reviewer #1**

It's my pleasure to review hess-2020-657 "Interaction of soil water and groundwater during the freezing-thawing cycle: field observations and numerical modeling" by Xie et al. The authors quantified the impact of freezing-induced groundwater migration and lateral groundwater inflow on soil moisture profile and groundwater level dynamics at site scale using the SHAW model. This is a very specific study at a single site, and the broad implication of this study to the relevant research community is unknown and needs to be justified. In addition, additional numerical experiments should be included to better quantify the impact of freezing-induced groundwater migration and lateral groundwater inflow, and additional descriptions of the measurements and methods are also necessary. According to these, a major revision is recommended. My comments are as follows.

Response: Thanks for your comments. We will introduce the broad implication of this study, add additional numerical experiments in the revision and give more detailed description of the measurements and methods in the revision.

Major Comments:

1. This paper describes a specific case of observing and simulating the impact of freezing-induced groundwater migration and lateral groundwater inflow on soil moisture profile and groundwater level dynamics at a single site with very shallow water level ranging from 90-143cm. I think this is a very special case for the frozen areas that the water level is generally much deeper. As such, the broad implication of this study to the relevant research community should be justified. In addition, the authors are suggested to include additional numerical simulations to investigate the impact of different water level depths.

Response: It is true that we have only one site with detailed measurements of time series of groundwater level and soil liquid water content, which is shown in the current study. However, this is not a special case. Shallow groundwater tables widely occur in river valleys and coastal regions, as well as in regions with large-scale topographic relief

(Gleeson et al., 2011; Fan et al., 2013). In our study area covered mainly by sand, numerical results show that when the water table depth equals 2.2 m, groundwater level is not influenced by freezing (Fig. 1d). We find the region with water table depth smaller than 2 m accounts for around one third of the total area of the current catchment. As shown in Fan et al.(2013), the regions with water table depth below 2 m account for around 31% of the global land area.

Thanks for the suggestion of adding additional numerical simulations. Because we found the extinction depth of freezing-induced groundwater table decline is 2.2 m in our study area, we will show the simulation results when initial water table depth equals 1.7m and 2.2 m (with and without lateral groundwater flow) in the revision, which are shown below (Figure 1).

[Figure]

Figure 1. The frost depth, groundwater level and total water content under four scenarios. (a-b) initial water table depth equals 1.7 m with a lateral flow rate of 1.03 mm/d (a) and without lateral flow (b); (c-d) initial water table depth equals 2.2 m with a lateral flow rate of 1.03 mm/d (c) and without lateral flow (d).

2. Related to comment #1, I think detailed descriptions of the study site and all the available measurements are necessary as these given in Jiang et al. 2017,2018 cited in this paper. I found that there are at least two experimental wells and several other kinds of wells shown in Jiang et al. 2017, 2018. Why only one experimental well is investigated in this study? What's the typical depth of groundwater level found across the whole Wudu lake catchment? How far is the Otak meteorological station from the monitoring site? What's the accuracy of precipitation and soil moisture measurements? Did the authors perform site-specific calibration for the 5TM sensor? And how accurate can the 5TM sensor measure the liquid water content under frozen condition? What's the typical vegetation and soil types? The measurements of soil texture and soil temperature are also suggested to include in the supplement.

Response: Thanks for your suggestion of providing detailed descriptions of the study site and all the available measurement. In the revision, we will add the typical vegetation (*Achnatherum splendens*), soil type (loamy sand), and range of water table depths across the whole Wudu lake catchment (from less than 0.5 m to more than 15 m), as well as measurements of soil texture at 12 depths.

Although we have another monitoring well (DK 1 as shown in Jiang et al., 2018), the water table depth is as deep as around 15 m. Due to the deep water table depth, there is no interaction of soil water and groundwater induced by freezing. All other wells shown on the map of water table elevation in Jiang et al. (2018) are domestic wells.

The Otak meteorological station, which is a national meteorological station, is around 35 km away from the study site. The accuracy of precipitation measurements is ±0.1 mm. The 5TM capacitance sensors, which measure the soil dielectric permittivity to represent liquid soil water content, have an accuracy of around ±2% volumetric water content (VWC). It has been reported that the 5TM sensor was accurate enough to measure the liquid water content under frozen condition (Yang et al., 2013; Xue et al., 2021). We have performed site-specific calibration for the sensors by the comparing the VWC measured by the 5TM sensors and by the gravimetric method.

3. The authors indicated that "we find snowfall did not infiltrate into the soil column due to the low permeability of frozen soil", which I think is questionable. If the permeability of frozen soil is so low that the snowmelt cannot infiltrate into the soil, how can the freezing-induced groundwater migration enter the soil column? What's the mechanism behind this? I am curious how the authors simulate the snow process? What's the accuracy of snowfall measurements and snowmelt simulations?

Response: Thanks for pointing out the problem of the sentence "we find snowfall did not infiltrate into the soil column due to the low permeability of frozen soil". After referring to several references (Iwata et al., 2008; Zhao et al., 2013; Mohammed et al., 2018), we realize that although infiltration of snowmelt would be impeded by the low permeability of frozen soil, the majority of snowmelt can be infiltrated into the frozen zone. Unfortunately, because we did not set the parameter representing "ponding for rainfall and snowmelt" correctly, our model results led to the wrong conclusion that snowfall could not infiltrate into the soil column. We have fixed this problem and will re-run all models in the revision.

In the meteorological station, the amount of snowfall is determined by weighting water equivalent of snowfall, which has an accuracy of ±0.1 mm. The total snowfall during the freezing-thawing period is found to be 11.7 mm. In the SHAW model, precipitation is assumed to be snow if the air temperature is below 0℃, and snow would be melted when the temperature is increased to above 0℃. However, snow could be easily blown away by wind in a bare ground before melting (Link and Marks, 1999; Zhao et al., 2013). After correcting the parameter representing "ponding for rainfall and snowmelt", we find the amount of infiltration from snowmelt equals 3.23 mm, which accounts for 28% of the total snowfall. Because the infiltration of snowmelt is limited, the wrong treatment on snowmelt does influence other conclusions drawn in the current study. We will re-calculate the water budget and update all figures in the revision.

Concerning the question of "If the permeability of frozen soil is so low that the snowmelt cannot infiltrate into the soil, how can the freezing-induced groundwater migration enter the soil column?", we want to clarify that due to cryosuction at the

freezing front, groundwater migrates through the unfrozen zone and gets frozen near the freezing front. As shown in Fig.5 in the manuscript, as the freezing front moves down, the total water content in the frozen zone changes little.

4. Detailed descriptions of how the authors determine the hydraulic parameters are necessary. Did the authors measure the soil texture and other relevant hydraulic parameters such as porosity, bulk density and saturated hydraulic conductivity? Why the saturated hydraulic conductivity estimated for the second layer (0.7-1.0 m) is so different from other two layers? How the authors determine the permeability of aquifer?

Response: We measured particle size by the Mastersizer 2000 instrument (Malvern Instruments, England) and the bulk density of the different layers in the study site with the cutting-ring method. The measured soil parameters are shown in the Table 1. Based on the contents of clay, silt and sand, the profile is divided into three layers. Based on the average contents of clay, silt and sand in each layer, initial estimates of hydraulic parameters ($\theta_r$, $\theta_s$, $\alpha$, $n$) are estimated by the Rosetta pedotransfer function (Schaap and Leij 1998; Zhang and Schaap 2017). We further calibrated the hydraulic parameters by fitting the simulated and measured soil water content.

Table 1 The measured soil texture

| Depth (cm) | Clay (%) | Silt (%) | Sand (%) | Bulk density(g/cm$^3$) |
|---|---|---|---|---|
| 10 | 1.5 | 9.4 | 89.1 | 1.638 |
| 20 | 1.6 | 9.5 | 88.9 | 1.673 |
| 30 | 1.5 | 9.5 | 89.0 | 1.628 |
| 40 | 2.0 | 9.5 | 88.5 | 1.672 |
| 50 | 2.5 | 9.1 | 88.3 | 1.655 |
| 60 | 2.7 | 9.7 | 87.6 | 1.613 |
| 70 | 5.5 | 13.5 | 81.0 | 1.562 |
| 80 | 6.4 | 11.2 | 82.4 | 1.518 |
| 90 | 6.1 | 9.2 | 84.7 | 1.549 |
| 100 | 7.7 | 10.4 | 81.9 | 1.598 |
| 110 | 3.0 | 9.4 | 87.6 | 1.652 |
| 120 | 1.7 | 9.7 | 88.6 | 1.733 |

We also measured the saturated hydraulic conductivity of soil samples from each layer by HYPROP (www.metergroup.com/environment/products/hyprop-2/). The soil

samples with low clay content above 70 cm and below 100 cm were measured to be around 18.0 cm/h, while that with higher clay content at the depth ranging between 70 and 100 cm is measured to be 0.8 cm/h. The different saturated hydraulic conductivity is caused by the slight difference in clay and silt. The permeability of aquifer is the same as saturated hydraulic conductivity of soils below 100 cm.

5. Detailed descriptions of the SHAW model and its implementation are necessary. For instance, how the model compute the permeability of frozen soil? How the authors include the lateral groundwater inflow into the SHAW model? How the authors determine the temperature at the lower boundary? What are the state variables need to be determined before the simulations? What's the time step of simulations? How the authors consider the impact of vegetation processes?

Response: We will give more detailed description of the SHAW model in the revision. Here, we briefly answer the questions.

The permeability of both unfrozen soil and frozen soil is computed by the van Genuchten and Mualem equation. However, when the porosity of frozen soil is decreased to 0.13, the permeability is assumed to be zero.

The lateral groundwater inflow is added to the saturated zone of the 1D soil column. Specifically, we assign a constant horizontal hydraulic gradient at one node within the saturated zone. The SHAW model calculates the lateral flow rate based on the assigned horizontal hydraulic gradient and saturated hydraulic conductivity.

For scenarios A and B which have the length of the soil column is 155 cm), we use the measured temperature in 150 cm at the lower boundary. When there is no temperature measurement at the lower boundary, the temperature at the lower boundary can be estimated the by the force-restore approach, which is shown by the following expression (Hirota et al., 2002):

$$\left(1+\frac{2z}{d_d}\right)\frac{\partial T}{\partial t} = \frac{2}{C_S d_d}G - \omega\left(T - T_{AVG}\right) \tag{1}$$

where z is the depth [L] below the surface, $\omega$ is the frequency [$\Theta^{-1}$] of fluctuation period, $d_d$ is damping depth [L] corresponding to $\omega$, which is expressed as $d_d = \left( \dfrac{2k_T}{C_S \omega} \right)^{1/2}$, $k_s$ is the is volumetric heat capacity of soil [$M\,L^{-2}\,T^{-2}\,\Theta^{-1}$], $k_T$ is thermal conductivity of soil [$M\,L^2\,T^{-3}\,\Theta^{-1}$], and $T_{AVG}$ is the average annual air temperature. Equation 1 is embedded in the SHAW model.

Because we find the soil temperature estimated at 150 cm by using equation 1 deviate slightly from soil temperature measured at 150 cm, to insure the accuracy of model calibration, we use the measured temperature at 150 cm to represent the bottom temperature for the base case model. After model calibration, we change the length of the soil column into 200 cm to account for the scenarios with deeper water table depths, and use equation 1 to obtain temperature at the lower boundary. In the revision, we will extend the length of the soil column into 250 cm to account for more scenarios of freezing-induced groundwater migration.

The stable variables of the model include soil temperature and soil water content. We use the initial conditions of soil water content and soil temperature on 29 OCT 2015 for spin-up, which is run for 30 days to obtain the initial conditions before the start of freezing (28 NOV 2015). The time step of simulations is one hour.

Because the grass in our study site fades during the freezing and thawing stages, we don't consider the influence of plants in the model.

6. Four numerical experiments are conducted to investigate the impact of soil heterogeneity and lateral groundwater on the simulations. I do not find the necessary to quantify the impact of soil heterogeneity. Instead, I think the authors can consider following additional experiments such as simulations without impact of groundwater, simulations with deeper water level depth, and simulations with changing rates of lateral groundwater inflow. It's not clear why the authors fix the rate of lateral groundwater inflow.

Response: Thanks for the suggestions, we will add some additional simulations with deeper water table depth (Figure 1 in the current file), and some simulations without impact of groundwater.

Although we have shown the water level fluctuations under different rates of lateral groundwater inflow (Figure 7 in the manuscript), we will add plots to show how different rates of lateral groundwater inflow would influence total water content and frost depth. Because lateral groundwater flow is mainly controlled by regional-scale water table undulation, we assume that rate of lateral groundwater flow is constant during the whole freezing-thawing period.

Because the vertical distribution of soil water content in the profile is influenced by the low-permeability layer from 70 to 100 cm, we prefer to keep the scenarios with homogeneous soil to explain how a middle layer with low-permeability influence freezing-induced groundwater migration as well as evaporation during the freezing-thawing stage .

Minor Comments:

1. The authors are suggested to merge Figures 4 and 5, and the info of precipitation is suggested to include in the figure. The scale for the temperature can be set at 10~-20.

Response: Thanks for your suggestions. We will add the information of precipitation into the figure. However, we respectfully disagree to merge figures 4 and 5. Figure 4 shows the sensitivity of measured liquid water content to temperature drop from above 0℃ to below 0℃, and well demonstrates that 5TM sensors can be used to monitor liquid water content during the freezing period.

2. Why there is not increase found for the simulation of total water content at 10 cm as shown in Figure 5?

Response: The low total water content in the shallow part (from the surface to 20 cm) of the soil profile is caused by the low initial soil water content at these depths as well as the long distance away from the water table.

3. It's suggested to plot the measured frost depth and WTD in all the subplots of Figure 6. In addition, how the authors determine the frost depth?

Response: We will add the measured frost depth and WTD into all subplots as suggested. The measured frost depth is determined by the dates of start of freezing and end of thawing as shown in the figure below.

[Figure]

Figure 2.The dates of start of freezing and end of thawing to determine frost depth

4. For all the simulations, it's suggested to show how they affect both soil liquid water content and temperature simulations, as well as to list the corresponding error statistics.

Response: The simulated liquid water content and temperature for the four scenarios are shown in Figure3 and 4. We will incorporate these plots in the revision.

The error statistics for all of the four scenarios are listed in Tables 2 and 3. We will add the results in the revision.

Table 2 The RMSEs of simulated liquid water content under different scenarios

| Parameters | 10 cm | 20 cm | 30 cm | 50 cm | 70cm |
|---|---|---|---|---|---|
| Scenario A | 0.0213 | 0.0205 | 0.0289 | 0.0175 | 0.0231 |
| Scenario B | 0.0237 | 0.0300 | 0.0322 | 0.0321 | 0.0761 |
| Scenario C | 0.0224 | 0.0332 | 0.0297 | 0.0352 | 0.0425 |
| Scenario D | 0.0253 | 0.0361 | 0.0273 | 0.0497 | 0.0825 |

Table 3 The RMSEs of simulated soil temperature under different scenarios

| Parameters | 10 cm | 20 cm | 30 cm | 50 cm | 70cm |
|---|---|---|---|---|---|
| Scenario A | 0.95 | 0.67 | 0.69 | 0.41 | 0.37 |
| Scenario B | 1.06 | 0.78 | 0.79 | 0.47 | 0.41 |
| Scenario C | 1.24 | 0.91 | 0.88 | 0.55 | 0.54 |
| Scenario D | 1.30 | 1.03 | 0.99 | 0.68 | 0.64 |

[Figure]

Figure 3 The frost depth, groundwater level and liquid water content under the four scenarios. (a) Scenario A, heterogeneous soil profile with a lateral flow rate of 1.03 mm/d; (b) Scenario B, homogeneous soil profile with a lateral flow rate of 1.03 mm/d; (c) Scenario C, heterogeneous soil profile without lateral inflow; (d) Scenario D, homogeneous soil profile without lateral inflow.

[Figure]

Figure 4 The frost depth, groundwater level and temperature under the four scenarios. (a) Scenario A, heterogeneous soil profile with a lateral flow rate of 1.03 mm/d; (b) Scenario B, homogeneous soil profile with a lateral flow rate of 1.03 mm/d; (c) Scenario C, heterogeneous soil profile without lateral inflow; (d) Scenario D, homogeneous soil profile without lateral inflow.

5. For the description of soil evaporation in Section 3.3, can the authors provide validation data? If not, I don't find the necessary to include this subsection. Instead, the impact of different numerical experiments on both soil liquid water content and temperature simulations can be shown in detail.

Response: Thanks for your suggestions. It is a pity that we do not have measured data to validate simulated results of soil evaporation, so we agree to delete subsection 3.3. We will incorporate plots of simulated soil liquid water content and simulated temperature in the revision.

6. For the Table S1, it's suggested to remove 110cm and 150cm since there are not measurements recorded for these two depths. Besides, it's suggested to add the measured soil temperature and lateral groundwater inflow in the supplement.

Response: Thanks for your suggestions. We will remove 110 cm and 150 cm in Table S1. In fact, the measured soil temperature is already listed in Table S2 of supplement material.

We want to clarify that we don't have measured lateral groundwater inflow data. The rate of lateral groundwater inflow was initially estimated by Jiang et al. (2017) based on water table fluctuations in the unfrozen period. In the current study, we estimated the rate of lateral groundwater inflow by fitting the simulated water level and measured water level during the freezing-thawing stage.

**References**

Fan, Y., Li, H., and Miguez-Macho, G.: Global patterns of groundwater table depth, Science, 339, 940-943, 10.1126/science.1229881, 2013.

Gleeson, T., Marklund, L., Smith, L., and Manning, A. H.: Classifying the water table at regional to continental scales, Geophysical Research Letters, 38, L05401, 10.1029/2010gl046427, 2011.

Hirota, T.: An extension of the force-restore method to estimating soil temperature at depth and evaluation for frozen soils under snow, Journal of Geophysical Research, 107, 4767, 10.1029/2001jd001280, 2002.

Iwata, Y., Hayashi, M., and Hirota, T.: Effects of snow cover on soil heat flux and freeze-thaw processes. Journal of Agriculture Meteorology, 64(4):301-309, 10.2480/agrmet.64.4.12, 2008.

Link T., Marks D., Distributed simulation of snowcover mass- and energy-balance in the boreal forest, Hydrological Processes, 13(14-15): 2439-2452, 10.1002/(SICI)1099-1085(199910)13:14/15<2439::AID-HYP866>3.0.CO, 2015.

Mohammed A. A., Kurylyk B. L., Cey E. E., Hayashi, M.: Snowmelt infiltration and macropore flow in frozen soils: overview, knowledge gaps, and a conceptual framework, 10.2136/vzj2018.04.0084, Vadose Zone Journal, 17(1), 2018.

Schaap, M. G., and Leij, F. J.: Database-Related Accuracy and Uncertainty of Pedotransfer Functions, Soil science, 163, 765-779, 10.1097/00010694-199810000-00001, 1998.

Xue, K., Wen, Z., Zhu, Z., Wang, D., and Zhang, M.: An experimental study of the relationship between the matric potential, unfrozen water, and segregated ice of saturated freezing soil, Bulletin of Engineering Geology and the Environment, 80(3), 10.1007/s10064-020-02052-x, 2021.

Yang, K., Qin, J., Zhao, L., Chen, Y., Tang, W., Han, M., Chen, Z., Lv, N., Ding, B., Wu, H., and Lin, C.: A multiscale soil moisture and freeze-thaw monitoring network on the third pole, Bulletin of the American Meteorological Society, 94(12), 1907-1916, 10.1175/BAMS-D-12-00203.1, 2013.

Zhang, Y, Schaap, M. G., and Marcel, G.: Weighted recalibration of the Rosetta pedotransfer model with improved estimates of hydraulic parameter distributions and summary statistics (Rosetta3), Journal of Hydrology, 547, 39-53, 10.1016/j.jhydrol.2017.01.004, 2017.

Zhao, Y., Huang, M., Horton, R., Liu, F., Peth, S., and Horn, R.: Influence of winter grazing on water and heat flow in seasonally frozen soil of Inner Mongolia, Vadose Zone Journal, 12(1), 10.2136/vzj2012.0059, 2013.

---

## Author Comment (AC2)

**Response to Reviewer #2**

The work presented by Xie et al. (2021) investigated the interaction of soil water and groundwater mainly via the lateral groundwater flow and freezing or thawing induced water migration during the freezing-thawing cycle in a semi-arid region with shallow groundwater. They conducted field observations and numerical experiments and further analyzed the water budget components. The role of lateral groundwater flow and the freezing-thawing process was demonstrated important in the tested area. I found this work is interesting while there are some concerns about the current version of the manuscript necessary to be addressed from my perspective.

First, the existence of freezing-induced water gain and lateral groundwater flow is mostly postulated from the observations and not directly measured. This renders that you have to demonstrate the reliability and uncertainties of your observations (e.g., liquid water content, the occurrence of thawed water infiltration, frost depth, …).

Response: Thanks for your comments. We will add more details of the reliability of our observations in the revision. The 5TM sensors, which measure the soil dielectric permittivity to represent liquid soil water content, have an accuracy of ±2% volumetric water content. It has been reported that the 5TM sensor was accurate enough to measure the liquid water content under frozen condition (Yang et al., 2013; Zhang et al., 2019; Xue et al., 2021). The 5TM sensors have an accuracy of ±1℃ for soil temperature, which is used to infer the frost depth. In the manuscript, Figure 4 also shows that the measured liquid water content is sensitive to the temperature drop from above 0℃ to below 0℃, which well demonstrates that 5TM sensors can be used to monitor liquid water content and temperature during the freezing period. Because there are increases in liquid soil water content during the thawing stage, we believe that the increased liquid soil water content is caused by the infiltration of thawed water.

Second, as most of the analysis part is based on the SHAW model simulations. I think the authors should put a bit more words on the SHAW model setup (e.g., bottom

boundary condition settings, how groundwater is considered), model performance, and uncertainty (e.g., simulation of freezing/thawing dates, statistical performance). Thus, I suggest more dedicated efforts should be made before its publication in the HESS journal.

Response: Thanks for your suggestions. We will add more detailed description of the SHAW model setup, model performance and uncertainty in the revision.

My specific comments are as follows:

Abstract:

Line 22-23: I notice that in your figures (Figure 4) the unit of soil water content is cm3/cm3, please keep consistent.

Response: Thanks for your suggestions; we will add the unit cm3/cm3 in the revision.

1 Introduction:

Line 44: "… and further decrease the hydraulic conductivity of frozen soils" please explain or clarify.

Response: Thanks for your comments. When there is groundwater migrated into the freezing front, the porosity might be decreased to be as low as 0.13. In this case, the permeability is considered to be 0 in the SHAW model. Therefore, we wrote the sentence "Soil freezing also generated cryosuction, which causes migration of water from the unfrozen zone to the freezing front and further decrease the hydraulic conductivity of frozen soils". Now we realize that it is not appropriate to add "… and further decrease the hydraulic conductivity of frozen soils" before mentioning freezing-induced groundwater migration. We will modify it in the revision.

Line 99-103: these sentence does not belong here, I think it should better be in the Introduction part.

Response: Thanks for your suggestions. We will consider to put them in a proper place in the revision.

2 Method

Please clearly present the soil texture (fraction of sand, clay, silt, organic matter) information.

Response: We have measured particle size by the Mastersizer 2000 instrument (Malvern Instruments, England), the measured fraction of sand, clay, silt shown in the Table 1. However, we have not measured the organic matter and did not consider the influence of organic matter in our model.

Table 1 The measured fraction of sand, clay, silt

| Depth (cm) | Clay (%) | Silt (%) | Sand (%) |
|---|---|---|---|
| 10 | 1.5 | 9.4 | 89.1 |
| 20 | 1.6 | 9.5 | 88.9 |
| 30 | 1.5 | 9.5 | 89.0 |
| 40 | 2.0 | 9.5 | 88.5 |
| 50 | 2.5 | 9.1 | 88.3 |
| 60 | 2.7 | 9.7 | 87.6 |
| 70 | 5.5 | 13.5 | 81.0 |
| 80 | 6.4 | 11.2 | 82.4 |
| 90 | 6.1 | 9.2 | 84.7 |
| 100 | 7.7 | 10.4 | 81.9 |
| 110 | 3.0 | 9.4 | 87.6 |
| 120 | 1.7 | 9.7 | 88.6 |

Are the 5TM sensors calibrated on this site? What about the measuring accuracy of 5TM sensors regarding the soil moisture and temperature?

Response: We have performed site-specific calibration for the sensors by the comparing the volumetric water content measured by the 5TM sensors and by the gravimetric method. The 5TM sensors have an accuracy of around ±2% for the volumetric water content and around ±1℃ for the temperature.

Line 112-114: From my understanding, Figure 1 cannot directly tell us that the increase of groundwater table depth is due to the freezing-induced water migration. Please carefully rephrase this sentence.

Response: Thanks for pointing out this. We totally agree that "Figure 1 cannot directly tell us that the increase of groundwater table depth is due to the freezing-induced water migration". We will modify it into "As shown in Fig. 1, water table depth changes from 115 cm on 28 NOV 2015 to 143 cm on 29 JAN 2016, which corresponds to the stage with an increasing frost depth."

2.3 Model inputs

Line 174: "… soil column is set to be 155 cm." I can understand that you want to use the measured soil temperature (150 cm) as the bottom boundary condition. While for the other numerical scenarios, you use 200cm for the soil column. I think either you keep all the numerical scenarios as 200cm, or you should explain a little bit the potential uncertainties regarding the results.

Response: It is true that it is appealing to use a soil column with the same length for all simulations. However, we do not have soil temperature measurement deeper than 150 cm below surface. When there is no soil temperature measurement at the lower boundary, the temperature can be estimated the by the force-restore approach, which is shown by the following expression (Hirota et al., 2002):

$$\left(1+\frac{2z}{d_d}\right)\frac{\partial T}{\partial t}=\frac{2}{C_S d_d}G-\omega\left(T-T_{AVG}\right) \tag{1}$$

where z is the depth [L] below the surface, $\omega$ is the frequency [$\Theta^{-1}$] of fluctuation period, $d_d$ is damping depth [L] corresponding to $\omega$, which is expressed as $d_d=\left(\frac{2k_T}{C_S \omega}\right)^{1/2}$, $k_s$ is the is volumetric heat capacity of soil [M L$^{-2}$ T$^{-2}$ $\Theta^{-1}$], $k_T$ is thermal conductivity of soil [M L$^2$ T$^{-3}$ $\Theta^{-1}$], and $T_{AVG}$ is the average annual air temperature. Equation 1 is embedded in the SHAW model. Unfortunately, we find the estimated soil temperature at 150 cm deviates slightly from the observed values.

Here, we briefly introduce how we decide to treat the bottom boundary condition. We will use a soil column with a length of 150 cm as the base case for model calibration and calculating water budget. Because we find numerically that groundwater level is not influenced by freezing when the water table depth equals 220 cm, we will use a soil column with a length of 250 cm in all other simulations for sensitivity analysis (initial water table depth equals 220 cm and 170 cm).

Line 176: Please explain in more detail about the lateral groundwater flow in SHAW. And how did you calibrate the lateral groundwater flow as 1.03mm/d?

Response: The lateral groundwater inflow is added to the saturated zone of the 1D soil column. Specifically, we assign a constant horizontal hydraulic gradient at one node within the saturated zone. The SHAW model calculates the lateral flow rate based on the assigned horizontal hydraulic gradient and saturated hydraulic conductivity. The rate of lateral groundwater inflow was estimated by Jiang et al. (2017) to range from 0.96 mm/d to 1.16 mm/d based on water table fluctuations in the unfrozen period. In the current study, the rate of lateral groundwater inflow is estimated by fitting the simulated and measured water level during the freezing-thawing stage. We find a lateral inflow rate of 1.03 mm/d leads to the best fit of water table depth (with the lowest RMSE).

Line 187: For the bottom boundary conditions, I have a sense that the setting of the bottom boundary condition can affect the simulations. Please explain why you set the bottom as the no-flux boundary condition. Is it the more realistic condition for that site or for the better simulations?

Response: To account for the effect of groundwater, a saturated zone should be included in the soil column. As we pointed out in the manuscript, a constant head lower boundary condition would overestimate freezing-induced groundwater migration toward the freezing front. Because the water level fluctuations are not independent from the atmospheric conditions, a specified-head lower boundary condition is not justified. Therefore, a closed soil column with a no-flow lower

boundary condition is the best choice to identify how the groundwater level responds to freezing-induced water migration, lateral groundwater flow, and infiltration of thawed water. This lower boundary condition leads to an accurate estimation of freezing-induced groundwater level decline as well as freezing-induced water gain in the frozen zone.

Line 204: For presenting the different simulation scenarios (A, B, C, D), I would suggest that you include them as a table, listing the main difference among all the simulation scenarios (or numerical experiments).

Response: Thanks for your suggestions. We will include a table to show the main difference among all the simulation scenarios in the revision. We will also add more numerical experiments in the revision.

Table 2: I think the row with "Calibrated parameters" should be below the row with "Initial parameters".

Response: We will exchange positions of the two rows in the revision.

3 Results and discussion

Figure 4: the scale of soil temperature should be finer (e.g., [-5, 10] oC) for Figure 4d, e, f. In addition, I think there also be freezing or thawing periods for 90cm. Please zoom in and clarify.

Response: Thanks for your suggestions. We will modify the scales of soil temperature in Figure 4. According to our measurements, the temperature of 90 cm never drop below 0℃, therefore, there is no freezing or thawing periods in the 90cm. We will zoom in the scale of temperature in the revision.

Line 248: please explain how to define the "occurrence of thawed water infiltration".

Response: In the manuscript, we wrote "From 29 JAN, there is a trend of rising water level, which is one month earlier than the occurrence of thawed water infiltration during the thawing stage." If there is no lateral groundwater inflow, the only possible cause of groundwater level rise is infiltration of thawed water. However, as we

observed, the water level rises before the thawing stage. Therefore, we interpret that the water level rise before the thawing stage is caused by lateral groundwater inflow.

Figure 5: I suggest that the statistical performance should be added here to demonstrate the capability of the SHAW model in simulating soil moisture and temperature.

Response: We agree and will add the statistical performance on the Figure 5 in the revision.

Section 3.2: please also add some text to describe how the model captures the observed freezing or thawing dates.

Response: Thanks for your suggestions. We will add some text to describe how the model captures the observed freezing or thawing dates in the revision.

Figure 6: for better comparison, please present the observed frost depth and water table depth for all four subplots. In addition, how is the frost depth measured?

Response: The measured frost depth is determined by the dates of start of freezing and end of thawing as shown in Figure 4 in the manuscript. We will add the observed frost depth and water table depth for all the subplots in the revision.

Table 3: please clarify the frozen zone and how you calculate TWC.

Response: In Table 3, the frozen zone represents the zone from the surface to the maximum frost depth. The mean TWC is calculated by total water content in the frozen zone ($S_{FZ}$) divided by the maximum frost depth (FD), while the $\Delta TWC$ is calculated by total water content change in the frozen zone ($\Delta S_{FZ}$) divided by the maximum frost depth (FD). We will explain in more detail in the revision.

Section 3.3: as you present two subplots in Figure 8, it is better to say what you want to say about the comparison here. The effect of snow can be clearly identified (Figure 8), the role or the amount of snowfall should be stated from the water budget closure perspective.

Response: Thanks for your suggestions. We will add the purpose of comparison in the revision. As pointed out by Reviewer #1, we did not correctly simulate the effect of snow in the current version. We have fixed this problem and find the amount of infiltration from snowmelt should be 3.23 mm. We will describe the effect of snow in detail in the revision.

4 Conclusions

I suggest adding some text describing that how well the SHAW model can capture the observed soil moisture, temperature, frost depth, and groundwater table depth.

Response: We will follow your suggestions in the revision.

Line 380: "a model is built…" should better be "a series of numerical experiments were set up to …" or something similar.

Response: Thanks for your suggestions. We will revise this sentence in the revision.

**References**

Hirota, T.: An extension of the force-restore method to estimating soil temperature at depth and evaluation for frozen soils under snow, Journal of Geophysical Research, 107, 4767, 10.1029/2001jd001280, 2002.

Jiang, X.-W., Sun, Z.-C., Zhao, K.-Y., Shi, F.-S., Wan, L., Wang, X.-S., and Shi, Z.-M.: A method for simultaneous estimation of groundwater evapotranspiration and inflow rates in the discharge area using seasonal water table fluctuations, Journal of hydrology, 548, 498-507, 10.1016/j.jhydrol.2017.03.026, 2017.

Xue, K., Wen, Z., Zhu, Z., Wang, D., and Zhang, M.: An experimental study of the relationship between the matric potential, unfrozen water, and segregated ice of saturated freezing soil, Bulletin of Engineering Geology and the Environment, 80(3), 10.1007/s10064-020-02052-x, 2021.

Yang, K., Qin, J., Zhao, L., Chen, Y., Tang, W., Han, M., Chen, Z., Lv, N., Ding, B., Wu, H., and Lin, C.: A multiscale soil moisture and freeze-thaw monitoring network on the third pole, Bulletin of the American Meteorological Society, 94(12), 1907-1916, 10.1175/BAMS-D-12-00203.1, 2013.

Zhang, Z., Wang, W., Gong, C., Wang, Z., Duan, L., Yeh, T. c. J., and Yu, P.: Evaporation from seasonally frozen bare and vegetated ground at various groundwater table depths in the Ordos Basin, Northwest China, Hydrological Processes, 10.1002/hyp.13404, 2019.

---

## Author Response (AR1)

**Cover Letter**

Dear Editor,

Thanks for providing detailed reviews of our manuscript *Interaction of soil water and groundwater during the freezing-thawing cycle: field observations and numerical modeling*. Following the suggestions of the two reviewers, we have rewritten the majority of the manuscript.

A detailed response is attached in this file. If there is any problem with the revision, please let me know.

Thank you for your consideration.

Best regards,

Xiao-Wei Jiang
Professor of Hydrology
E-mail: jxw@cugb.edu.cn

**Responses to the Reviewers**

Reviewer #1

It's my pleasure to review hess-2020-657 "Interaction of soil water and groundwater during the freezing-thawing cycle: field observations and numerical modeling" by Xie et al. The authors quantified the impact of freezing-induced groundwater migration and lateral groundwater inflow on soil moisture profile and groundwater level dynamics at site scale using the SHAW model. This is a very specific study at a single site, and the broad implication of this study to the relevant research community is unknown and needs to be justified. In addition, additional numerical experiments should be included to better quantify the impact of freezing-induced groundwater migration and lateral groundwater inflow, and additional descriptions of the measurements and methods are also necessary. According to these, a major revision is recommended. My comments are as follows.

Response: Thanks for your comments. We agree that our study was concentrated too much on the specific study site. To avoid this problem, we added more simulation results under a series of water table depths and lateral groundwater inflow rates in subsection 3.4 and the broad implication of this study in the last paragraph of the Conclusions part (lines 434-444). We also added more details of the measurements in subsection 2.2 and simulation methods in subsection 2.3.

Major Comments:

1. This paper describes a specific case of observing and simulating the impact of freezing-induced groundwater migration and lateral groundwater inflow on soil moisture profile and groundwater level dynamics at a single site with very shallow water level ranging from 90-143cm. I think this is a very special case for the frozen areas that the water level is generally much deeper. As such, the broad implication of this study to the relevant research community should be justified. In addition, the authors are suggested to include additional numerical simulations to investigate the impact of different water level depths.

Response:

In our study area covered mainly by sand, numerical results show that when the water table depth equals 2.2 m, groundwater level is not influenced by freezing. Because areas with water table depth shallower than 2 m account for around one third of the total catchment (Jiang et al., 2018) and regions with water table depth shallower than 2 m account for around 31% of the global land area (Fan et al., 2013), the involvement of groundwater in freezing-induced water redistribution could be widespread. Although the threshold water table depth in other study areas with different climate conditions remains unknown, our study well demonstrated the necessity of properly characterizing freezing-induced water table fluctuations to quantify freezing-induced groundwater

migration and its effect on engineering problems and ecological processes in cold regions. The broad implication of this study has been added in the last paragraph of the Conclusions part (lines 434-444)

We have added Figure 8 and Table 5 in subsection 3.4 to analyze the impact of freezing-induced groundwater migration under four initial water table depths (120 cm, 170 cm, 220 cm and 250 cm) and three rates of lateral groundwater inflow (0, 0.51 mm/d and 1.03mm/d). We also built three scenarios with water table depths fixed at 120 cm, 170 cm, and 220 cm. The specific settings of the 13 scenarios are shown in Table 3 in the manuscript.

2. Related to comment #1, I think detailed descriptions of the study site and all the available measurements are necessary as these given in Jiang et al. 2017,2018 cited in this paper. I found that there are at least two experimental wells and several other kinds of wells shown in Jiang et al. 2017, 2018. Why only one experimental well is investigated in this study? What's the typical depth of groundwater level found across the whole Wudu lake catchment? How far is the Otak meteorological station from the monitoring site? What's the accuracy of precipitation and soil moisture measurements? Did the authors perform site-specific calibration for the 5TM sensor? And how accurate can the 5TM sensor measure the liquid water content under frozen condition? What's the typical vegetation and soil types? The measurements of soil texture and soil temperature are also suggested to include in the supplement.

Response: In the revision, we have added the details of the Wudu lake catchment and the monitoring site in subsection 2.1, including the typical vegetation (*Achnatherum splendens*), soil type (loamy sand), and range of water table depths across the whole catchment (from less than 0.5 m to around 26 m), as well as soil texture measurements at 12 depths (Table 1 in the revision).

Although we have another monitoring well (DK 1 as shown in Jiang et al., 2018), the water table depth is as deep as around 15 m. Due to the deep water table depth, there is no interaction between soil water and groundwater induced by freezing. All other wells shown on the map of water table elevation in Jiang et al. (2018) are domestic wells for drinking. Therefore, only the monitoring DK2 is used in the current study.

We have added more details of the Otak meteorological station, which is a national meteorological station 35 km away from the study site. The accuracy of precipitation measurements is ±0.1 mm. The descriptions can be found in subsection 2.1 of the revision (lines 94-102).

The 5TM sensors, which measure the soil dielectric permittivity to represent liquid soil water content, have an accuracy of around ±2% volumetric water content (VWC) and a resolution of 0.1% VWC. We have performed site-specific calibration for the sensors by the comparing the VWC measured by the 5TM sensors and by the gravimetric method. We have added more detailed descriptions about the 5TM sensors in subsection 2.2 (lines 119-124).

3. The authors indicated that "we find snowfall did not infiltrate into the soil column due to the low permeability of frozen soil", which I think is questionable. If the permeability of frozen soil is so low that the snowmelt cannot infiltrate into the soil, how can the freezing-induced groundwater migration enter the soil column? What's the mechanism behind this? I am curious how the authors simulate the snow process? What's the accuracy of snowfall measurements and snowmelt simulations?

Response: Thanks for pointing out the problem of the sentence "we find snowfall did not infiltrate into the soil column due to the low permeability of frozen soil". After referring to several references (Iwata et al., 2008; Zhao et al., 2013; Mohammed et al., 2018), we realize that although infiltration of snowmelt would be impeded by the low permeability of frozen soil, the majority of snowmelt can be infiltrated into the frozen zone. Unfortunately, because we did not set the parameter representing "ponding for rainfall and snowmelt" correctly, our model results led to the wrong conclusion that "snowfall could not infiltrate into the soil column". After correcting the parameter representing "ponding for rainfall and snowmelt", we find the amount of infiltration from snowmelt equals 3.23 mm, which accounts for 28% of the total snowfall. We have updated all figures and re-calculated the water budget by including infiltration from snowmelt. We also added some description on how SHAW simulate the snow process in the subsection 2.3 of the revision (lines 190-199).

Concerning the question of "If the permeability of frozen soil is so low that the snowmelt cannot infiltrate into the soil, how can the freezing-induced groundwater migration enter the soil column?", we want to clarify that due to cryosuction at the freezing front, groundwater migrates through the unfrozen zone which lies above the water table and below the freezing front, and gets frozen near the freezing front. However, as the freezing front goes deeper, the ice content and total water content in the frozen zone above the freezing front change little (Figures 6 and 8).

4. Detailed descriptions of how the authors determine the hydraulic parameters are necessary. Did the authors measure the soil texture and other relevant hydraulic parameters such as porosity, bulk density and saturated hydraulic conductivity? Why the saturated hydraulic conductivity estimated for the second layer (0.7-1.0 m) is so different from other two layers? How the authors determine the permeability of aquifer?

Response: We measured particle size by the Mastersizer 2000 instrument (Malvern Instruments, England) and the bulk density with the cutting-ring method, the results of which are listed in Table 1. Based on the average contents of clay, silt and sand in each of the three layers, initial estimates of hydraulic parameters ($\theta_r$, $\theta_s$, $\alpha$, $n$) are estimated by the Rosetta pedotransfer function (Schaap and Leij 1998; Zhang and Schaap 2017). We further calibrated the hydraulic parameters by fitting the simulated and measured soil water content. The details of soil parameter measurement have been added in subsection 2.2 (lines 125-131).

We measured the saturated hydraulic conductivity of soil samples from each layer by HYPROP (www.metergroup.com/environment/products/hyprop-2/). The soil samples with low clay content above 70 cm and below 100 cm were measured to be around 18.0 cm/h, while that with higher clay content at the depth ranging between 70 and 100 cm is measured to be 0.8 cm/h. The different saturated hydraulic conductivity is caused by the slight difference in clay and silt. The hydraulic conductivity of aquifer is the same to the saturated hydraulic conductivity of the third layer. We have added the descriptions of how the authors determine the hydraulic parameters in subsection 2.4 (lines 228-235).

5. Detailed descriptions of the SHAW model and its implementation are necessary. For instance, how the model compute the permeability of frozen soil? How the authors include the lateral groundwater inflow into the SHAW model? How the authors determine the temperature at the lower boundary? What are the state variables need to be determined before the simulations? What's the time step of simulations? How the authors consider the impact of vegetation processes?

Response: We have added more detailed description of the SHAW model in subsection 2.3 in the revision (lines 160-166, lines 170-172, and lines 187-200). The hydraulic conductivity of both unfrozen soil and frozen soil is computed by the van Genuchten and Mualem equation. However, when the porosity of frozen soil is decreased to 0.13, the permeability is assumed to be zero. The details can be found in subsection 2.3 of the revision (lines 187-189).

The SHAW model calculates the lateral flow rate based on the assigned horizontal hydraulic gradient and saturated hydraulic conductivity. Specifically, a constant horizontal hydraulic gradient is assigned at one node within the saturated zone. This is described in subsection 2.3 of the revision (lines 169-171).

For the lower boundary condition, we have two different settings in the revision. For the base case model (field site model), we use the measured soil temperature at 150 cm as the bottom temperature boundary condition to insure the accuracy of model calibration. After model calibration, we change the length of the soil column into 250 cm to account for the scenarios with deeper water table depths, and use the force-restore approach (Hirota et al., 2002) embedded in the SHAW model to obtain temperature at the lower boundary. The details can be found in subsection 2.4 of the revision (lines 249-255).

The stable variables of the model include soil temperature and soil water content. We use the initial conditions of soil water content and soil temperature on 29 OCT 2015 for spin-up, which is run for 30 days to obtain the initial conditions before the start of freezing (28 NOV 2015). The time step of simulations is one hour. Because the grass in our study site fades during the freezing and thawing stages, we do not consider the influence of plants in the model. These contents are described in subsection 2.4 of the revision (lines 206-207 and lines 221-223).

6. Four numerical experiments are conducted to investigate the impact of soil heterogeneity and lateral groundwater on the simulations. I do not find the necessary to quantify the impact of soil heterogeneity. Instead, I think the authors can consider following additional experiments such as simulations without impact of groundwater, simulations with deeper water level depth, and simulations with changing rates of lateral groundwater inflow. It's not clear why the authors fix the rate of lateral groundwater inflow.

Response: Thanks for your suggestions. We have deleted the investigation of the impact of soil heterogeneity in the revision. We added Figure 8 and Table 5 in subsection 3.4 to analyze the impact of freezing-induced groundwater migration under four initial water table depths (120 cm, 170 cm, 220 cm and 250 cm) and three rates of lateral groundwater inflow (0, 0.51 mm/d and 1.03mm/d). We also built three scenarios with water table depths fixed at 120 cm, 170 cm, and 220 cm.

Minor Comments:

1. The authors are suggested to merge Figures 4 and 5, and the info of precipitation is suggested to include in the figure. The scale for the temperature can be set at 10~-20.

Response: Thanks for your suggestions. Figure 4 shows the sensitivity of measured liquid water content to temperature drop from above 0℃ to below 0℃, and well demonstrates that 5TM sensors can be used to monitor liquid water content during the freezing period. We respectfully disagree to merge figures 4 and 5. Moreover, because the amount of precipitation has been shown in Figure 3, we prefer not to show them again in Figure 4. The scale for temperature at 10 cm, 20 cm and 30 cm is 10~-20. Because of the much smaller temperature variations at 50 cm, 70 cm and 90 cm, we follow the suggestion of Reviewer 2 and set the scale to be 10~-5.

2. Why there is not increase found for the simulation of total water content at 10 cm as shown in Figure 5?

Response: The low total water content in the shallow part of the soil profile is caused by the low initial soil water content as well as the long distance away from the water table. As shown in Figure 8, when the initial water table depth increases, the zone with low total water content becomes thicker.

3. It is suggested to plot the measured frost depth and WTD in all the subplots of Figure 6. In addition, how the authors determine the frost depth?

Response: Thanks for the suggestion. We have added the measured frost depth in all subplots as suggested, and added the measured water table depth in all subplots with an initial water table

depth of 120 cm. The measured frost depth is determined by the dates of start of freezing and end of thawing as shown in Figure 4 in the manuscript.

4. For all the simulations, it is suggested to show how they affect both soil liquid water content and temperature simulations, as well as to list the corresponding error statistics.

Response: Thanks for the suggestion. To show how ice content, soil liquid water content and temperature simulations respond to freezing and thawing, we have added subplots of ice content, soil liquid water content and temperature in Figure 6. The RMSEs of soil liquid water content and soil temperature are listed in Figure 5. Because we already have 12 subplots in Figure 8, we only show the distribution of total water content.

5. For the description of soil evaporation in Section 3.3, can the authors provide validation data? If not, I don't find the necessary to include this subsection. Instead, the impact of different numerical experiments on both soil liquid water content and temperature simulations can be shown in detail.

Response: Thanks for your suggestion. We have deleted the subsection on soil evaporation because we have no measured data to validate the simulated results.

6. For the Table S1, it's suggested to remove 110cm and 150cm since there are not measurements recorded for these two depths. Besides, it's suggested to add the measured soil temperature and lateral groundwater inflow in the supplement.

Response: Thanks for your suggestions. We have removed 110 cm and 150 cm in Table S1. In fact, the measured soil temperature is already listed in Table S2 of supplement material.

We want to clarify that we do not have measured lateral groundwater inflow data. The rate of lateral groundwater inflow was initially estimated by Jiang et al. (2017) based on water table fluctuations in the unfrozen period. In the current study, we estimated the rate of lateral groundwater inflow by fitting the simulated water level and measured water level during the freezing-thawing stage.

**Reviewer #2**

The work presented by Xie et al. (2021) investigated the interaction of soil water and groundwater mainly via the lateral groundwater flow and freezing or thawing induced water migration during the freezing-thawing cycle in a semi-arid region with shallow groundwater. They conducted field observations and numerical experiments and further analyzed the water budget components. The role of lateral groundwater flow and the freezing-thawing process was demonstrated important in the tested area. I found this work is interesting while there are some concerns about the current version of the manuscript necessary to be addressed from my perspective.

First, the existence of freezing-induced water gain and lateral groundwater flow is mostly postulated from the observations and not directly measured. This renders that you have to demonstrate the reliability and uncertainties of your observations (e.g., liquid water content, the occurrence of thawed water infiltration, frost depth, …).

Response: Thanks for your comments. We have added more details of the accuracy of our observations in subsection 2.2 (lines 119-124) and the reliability of our observations in subsection 3.1 by explaining how liquid water content, total water content, ice content and temperature respond to freezing and thawing (lines 264-284).

Second, as most of the analysis part is based on the SHAW model simulations. I think the authors should put a bit more words on the SHAW model setup (e.g., bottom boundary condition settings, how groundwater is considered), model performance, and uncertainty (e.g., simulation of freezing/thawing dates, statistical performance). Thus, I suggest more dedicated efforts should be made before its publication in the HESS journal.

Response: Thanks for your suggestions. We have added more detailed description of model setup in subsection 2.4. As explained in the Introduction part and illustrated in Figure 1, the water table separating saturated and unsaturated zones is dynamic and can respond to freezing and thawing. The lower boundary condition of the field case model is explained in lines 207-208, while that of the 13 scenarios for sensitivity analysis is explained in lines 241-251.

We have added detailed description of model performance by including a new Figure 6 "The evolution of simulated soil temperature, ice content, liquid water content and total water content at the field site" and a new paragraph in lines 308-317.

We agree that our model has uncertainty. We have added the RMSEs of soil temperature and liquid water content in Figure 5. The simulated freezing/thawing dates slightly deviate the measured dates, which are listed below (Table S1). Because they are close enough, we prefer not to list the simulated dates in the text, but only show the measured freezing/thawing dates in all subplots of Figure 6. We have added a sentence "There is a good match between simulated and measured frost depth, the latter of which is determined by the dates of start of freezing and end of thawing as shown in Figure 4." in line 308-310.

Table S1 A comparison of measured and simulated dates of freezing and thawing

| Depth (cm) | The date of start of freezing | | The date of end of thawing | |
|---|---|---|---|---|
| | Measured | Simulated | Measured | Simulated |
| 10 | 02 DEC | 02 DEC | 14 MAR | 14 MAR |
| 20 | 15 DEC | 13 DEC | 15 MAR | 15 MAR |
| 30 | 16 DEC | 19 DEC | 16 MAR | 16 MAR |
| 50 | 9 JAN | 7 JAN | 22 MAR | 22 MAR |
| 70 | 23 JAN | 24 JAN | 28 MAR | 26 MAR |

My specific comments are as follows:

Abstract:

Line 22-23: I notice that in your figures (Figure 4) the unit of soil water content is cm3/cm3, please keep consistent.

Response: We have deleted this sentence but we have added all the unit of soil water content, which is cm3/cm3 in the revision.

1 Introduction:

Line 44: "… and further decrease the hydraulic conductivity of frozen soils" please explain or clarify.

Response: Thanks for your comments. The sentence has been modified into "As a result of cryosuction generated by Soil freezing, water from the unfrozen zone would migrate to the freezing front." (lines 39-41).

Line 99-103: these sentence does not belong here, I think it should better be in the Introduction part.

Response: Thanks for your suggestions. We do find that these sentences are not appropriate to be put here and have deleted them in the revision.

2 Method

Please clearly present the soil texture (fraction of sand, clay, silt, organic matter) information.

Response: We have added the fraction of sand, clay, silt in Table 1. However, we have not measured the organic matter and did not consider the influence of organic matter in our model.

Are the 5TM sensors calibrated on this site? What about the measuring accuracy of 5TM sensors regarding the soil moisture and temperature?

Response: We have performed site-specific calibration for the sensors by the comparing the VWC measured by the 5TM sensors and by the gravimetric method. The 5TM sensors have an accuracy

of around ±2% for the volumetric water content and around ±1℃ for the temperature. These contents are described in subsection 2.3 of the revision (lines 119-124).

Line 112-114: From my understanding, Figure 1 cannot directly tell us that the increase of groundwater table depth is due to the freezing-induced water migration. Please carefully rephrase this sentence.

Response: Thanks for pointing out this. We totally agree. We have modified the sentence into "As shown in Fig. 2, water table depth increases from 115 cm on 28 NOV 2015 to 143 cm on 29 JAN 2016, which corresponds to the stage with an increasing frost depth." (lines 132-134)

2.3 Model inputs

Line 174: "… soil column is set to be 155 cm." I can understand that you want to use the measured soil temperature (150 cm) as the bottom boundary condition. While for the other numerical scenarios, you use 200 cm for the soil column. I think either you keep all the numerical scenarios as 200cm, or you should explain a little bit the potential uncertainties regarding the results.

Response: Thanks for your suggestions. We have re-run all models in the revision and used two different lower boundary conditions. For the base case model (field site model), we use the measured soil temperature at 150 cm as the bottom temperature boundary condition to insure the accuracy of model calibration. After model calibration, we change the length of the soil column into 250 cm to account for the scenarios with deeper water table depths, and use the force-restore approach (Hirota et al., 2002) embedded in the SHAW model to obtain temperature at the lower boundary. The details can be found in subsection 2.4 of the revision (lines 204-208 and lines 241-255).

Line 176: Please explain in more detail about the lateral groundwater flow in SHAW. And how did you calibrate the lateral groundwater flow as 1.03mm/d?

Response: The lateral groundwater inflow is added to the saturated zone of the 1D soil column. Specifically, we assign a constant horizontal hydraulic gradient at one node within the saturated zone. The SHAW model calculates the lateral flow rate based on the assigned horizontal hydraulic gradient and saturated hydraulic conductivity. The rate of lateral groundwater inflow was estimated by Jiang et al. (2017) to range from 0.96 mm/d to 1.16 mm/d based on water table fluctuations in the unfrozen period. In the current study, the rate of lateral groundwater inflow is estimated by fitting the simulated and measured water level during the freezing-thawing stage. We find a lateral inflow rate of 1.03 mm/d leads to the best fit of water table depth (with the lowest RMSE). The details can be found in lines 169-171 and lines 234-235.

Line 187: For the bottom boundary conditions, I have a sense that the setting of the bottom boundary condition can affect the simulations. Please explain why you set the bottom as the no-flux boundary condition. Is it the more realistic condition for that site or for the better simulations?

Response: To account for the effect of groundwater, a saturated zone should be included in the soil column. As we demonstrated in the manuscript, a constant head lower boundary condition would overestimate freezing-induced groundwater migration toward the freezing front. Because the water level fluctuations are not independent from the atmospheric conditions, a specified-head lower boundary condition is not justified. Therefore, a closed soil column with a no-flow lower boundary condition is the best choice to identify how the groundwater level responds to freezing-induced water migration, lateral groundwater flow, and infiltration of thawed water. This lower boundary condition leads to an accurate estimation of freezing-induced groundwater level decline as well as freezing-induced water gain in the frozen zone.

Line 204: For presenting the different simulation scenarios (A, B, C, D), I would suggest that you include them as a table, listing the main difference among all the simulation scenarios (or numerical experiments).

Response: Thanks for your suggestions. We have included Table 3 to show the main difference among all the simulation scenarios in the revision. We have added Figure 8 and Table 5 in subsection 3.4 to analyze the impact of freezing-induced groundwater migration under four initial water table depths (120 cm, 170 cm, 220 cm and 250 cm) and three rates of lateral groundwater inflow (0, 0.51 mm/d and 1.03mm/d). We also build three scenarios with water table depths fixed at 120 cm, 170 cm, and 220 cm.

Table 2: I think the row with "Calibrated parameters" should be below the row with "Initial parameters".

Response: We have deleted this table.

3 Results and discussion

Figure 4: the scale of soil temperature should be finer (e.g., [-5, 10] oC) for Figure 4d, e, f. In addition, I think there also be freezing or thawing periods for 90cm. Please zoom in and clarify.

Response: Thanks for your suggestions. We have modified the scales of soil temperature in Figure 4. According to our measurements, the temperature of 90 cm never drop below 0℃, therefore, there is no freezing or thawing periods in the 90cm, and we have zoomed in the scale of temperature.

Line 248: please explain how to define the "occurrence of thawed water infiltration".

Response: In the revision, we have changed the sentence "From 29 JAN, there is a trend of rising water level, which is one month earlier than the occurrence of thawed water infiltration during the thawing stage." into "the trend of rising water level from 29 JAN to 29 FEB, when frozen soil has not been thawed yet, is attributed to lateral groundwater inflow" (lines 291-292).

Figure 5: I suggest that the statistical performance should be added here to demonstrate the capability of the SHAW model in simulating soil moisture and temperature.

Response: We agree and have added the statistical performance (RMSE) on Figure 5 in the

revision.

Section 3.2: please also add some text to describe how the model captures the observed freezing or thawing dates.

Response: Thanks for your suggestions. We have added a sentence "There is a good match between simulated and measured frost depth, the latter of which is determined by the dates of start of freezing and end of thawing as shown in Figure 4." in line 308-310.

Figure 6: for better comparison, please present the observed frost depth and water table depth for all four subplots. In addition, how is the frost depth measured?

Response: We have added the measured frost depth in all subplots as suggested, and added the measured water table depth in all subplots with an initial water table depth of 120 cm. The measured frost depth is determined by the dates of start of freezing and end of thawing as shown in Figure 4 in the manuscript.

Table 3: please clarify the frozen zone and how you calculate TWC.

Response: In Table 5, the frozen zone represents the zone from the surface to the maximum frost depth. The mean TWC is calculated by total water content in the frozen zone ($S_{FZ}$) divided by the frost depth, while the $\Delta$TWC is calculated by total water content change in the frozen zone ($\Delta S_{FZ}$) divided by the maximum frost depth (FD). We have added how we calculate mean TWC and $\Delta$TWC in lines397-398.

Section 3.3: as you present two subplots in Figure 8, it is better to say what you want to say about the comparison here. The effect of snow can be clearly identified (Figure 8), the role or the amount of snowfall should be stated from the water budget closure perspective.

Response: Thanks for your suggestions. We have re-run the model by properly characterizing the behavior of snow and have deleted the evaporation of the homogenous case in the revision. Figure 7 in the revision has only one subplot to show the simulated evaporation process in our study site. We have updated the water budget calculation by including the snowmelt infiltration in subsect. 3.3.

4 Conclusions

I suggest adding some text describing that how well the SHAW model can capture the observed soil moisture, temperature, frost depth, and groundwater table depth.

Response: Thanks for your suggestions. We have add a sentence as "The observed fluctuating soil water contents and groundwater level induced by freezing and thawing are well reproduced by the calibrated model, which increases our understanding of water balance as well as freezing-induced water migration during the freezing-thawing cycle." (lines 422-425).

Line 380: "a model is built…" should better be "a series of numerical experiments were set up to …" or something similar.

Response: Thanks for your suggestions. We have rewritten the sentence as "Based on a series of models with different initial water table depths during the freezing-thawing cycle, …" (line 434).

References

Hirota, T.: An extension of the force-restore method to estimating soil temperature at depth and evaluation for frozen soils under snow, Journal of Geophysical Research, 107, 4767, 10.1029/2001jd001280, 2002.

Iwata, Y., Hayashi, M., and Hirota, T.: Effects of snow cover on soil heat flux and freeze-thaw processes. Journal of Agriculture Meteorology, 64(4):301-309, 10.2480/agrmet.64.4.12, 2008.

Jiang, X.-W., Sun, Z.-C., Zhao, K.-Y., Shi, F.-S., Wan, L., Wang, X.-S., and Shi, Z.-M.: A method for simultaneous estimation of groundwater evapotranspiration and inflow rates in the discharge area using seasonal water table fluctuations, Journal of hydrology, 548, 498-507, 10.1016/j.jhydrol.2017.03.026, 2017.

Jiang, X. W., Wan, L., Wang, X. S., Wang, D., and Zhao, K. Y.: A multi-method study of regional groundwater circulation in the Ordos Plateau, NW China, Hydrogeology Journal, 1657-1668, 10.1007/s10040-018-1731-4, 2018.

Link T., Marks D., Distributed simulation of snowcover mass- and energy-balance in the boreal forest, Hydrological Processes, 13(14-15): 2439-2452, 10.1002/(SICI)1099-1085(199910)13:14/15<2439::AID-HYP866>3.0.CO, 2015.

Mohammed A. A., Kurylyk B. L., Cey E. E., Hayashi, M.: Snowmelt infiltration and macropore flow in frozen soils: overview, knowledge gaps, and a conceptual framework, 10.2136/vzj2018.04.0084, Vadose Zone Journal, 17(1), 2018.

Xue, K., Wen, Z., Zhu, Z., Wang, D., and Zhang, M.: An experimental study of the relationship between the matric potential, unfrozen water, and segregated ice of saturated freezing soil, Bulletin of Engineering Geology and the Environment, 80(3), 10.1007/s10064-020-02052-x, 2021.

Yang, K., Qin, J., Zhao, L., Chen, Y., Tang, W., Han, M., Chen, Z., Lv, N., Ding, B., Wu, H., and Lin, C.: A multiscale soil moisture and freeze-thaw monitoring network on the third pole, Bulletin of the American Meteorological Society, 94(12), 1907-1916, 10.1175/BAMS-D-12-00203.1, 2013.

Zhang, Z., Wang, W., Gong, C., Wang, Z., Duan, L., Yeh, T. c. J., and Yu, P.: Evaporation from seasonally frozen bare and vegetated ground at various groundwater table depths in the Ordos Basin, Northwest China, Hydrological Processes, 10.1002/hyp.13404, 2019.

---

## Author Response (AR3)

**Cover Letter**

Dear Editor,

Thanks for providing detailed reviews of our manuscript *Interaction of soil water and groundwater during the freezing-thawing cycle: field observations and numerical modeling*. Following the suggestions of the two reviewers, we have revised the manuscript.

A detailed response is attached in this file. If there is any problem with the revision, please let me know.

Thank you for your consideration.

Best regards,

Xiao-Wei Jiang

Professor of Hydrology

E-mail: jxw@cugb.edu.cn

Responses to the Reviewers

Reviewer #1

It's my pleasure to review hess-2020-657 "Interaction of soil water and groundwater during the freezing thawing cycle: field observations and numerical modeling" by Xie et al. The authors have appropriately addressed my previous comments, and the paper can be accepted after addressing following minor comments.

Minor Comments:

1. The language can be further improved, and I note that many long sentences are often used, which can be divided into several short sentences.

Response: Thanks for your suggestion. We have divided many long sentences into short sentences. The language of the manuscript has also been improved.

2. For the description of implication, can the authors mention which parts of the world need to well demonstrate the freezing induced water table fluctuations?

Response: Seasonally frozen soils underlie approximately 24% of the Northern Hemisphere exposed land surface, the distribution of which is shown in Figure R1. As shown in Figure R2, regions with water table depth shallower than 2.5 m occur widely in the Northern Hemisphere. We found winter-time water table recession has been reported in such countries as Canada, China, Russia, Sweden and the USA. Although the number of field observations is still limited, we believe that the phenomenon could occur widely.

[Figure]

Figure R1 The distribution of permafrost and seasonally frozen ground in the Northern Hemisphere (Evans and Ge, 2017).

[Figure]

Figure R2 The distribution of global water table depth (Fan et al., 2013).

We have added the following sentences in the revision "Fan et al. (2013) reported that regions with water table depth shallower than 2 m account for around 31% of the global land area while Zhang et al. (2003) reported that seasonally frozen soils underlie approximately 24% of the Northern Hemisphere exposed land surface. Moreover, winter-time water table recession has been reported in such countries as Canada (van der Kamp et al., 2003;Ireson et al., 2013), China (Wu et al., 2016;Zhang et al., 2019), Russia (Vinnikov et al., 1996), Sweden (Stähli et al., 1999) and the USA (Drescher, 1955;Schneider, 1961;Daniel and Staricka, 2000), implying that the involvement of groundwater in freezing-induced water redistribution could be widespread."

3. Since the Otak station is 35 km away from the study site, I think the authors need to mention the potential impact of meteorological input uncertainties (e.g. spatial mismatch) on the simulations.

Response: Thank you for pointing out this problem. First of all, we want to emphasize that snowfall in the whole Ordos Plateau is limited and quite uniform. For example, in the winter from 2015 to 2016, the cumulative snowfalls in Otok and in Dongsheng are 11.3 mm and 10.7 mm, respectively. Note that the Dongsheng station is around 170 km away from the Otok station. Figure R3 shows the correlation between snowfall in Otok and snowfall in Dongsheng during 2013 and 2018. Therefore, we believe the uncertainty caused by snowfall is limited.

In our observation site, we installed a meteorological station in 2017. Figure R4 shows air temperature, relative humidity and wind speed from 1 NOV 2017 to 1 APR 2018 measured at the two meteorological stations. As shown in Figure R4 a, b and c, the maximum and minimum daily temperatures, and relative humidity at the two stations are close enough. The freezing index, i.e.,

the cumulative negative temperature, equals -885.8 ℃-days at the Otok meteorological station, and equals -897.4 ℃-days at our monitoring site. We believe that such a small difference in freezing index has limited impact on frost depth and freezing-induced groundwater migration.

[Figure]

Figure R3 A comparison between snowfalls in the Otok and Dongsheng meteorological stations from NOV 1 2013 to DEC 31 2018.

[Figure]

Figure R4 The comparison between the Otok meteorological station and DK2 meteorological station on (a) Maximum daily air temperature; (b) Minimum daily air temperature; (c) Relative humidity; (d) wind speed from 1 NOV 2017 to 1 APR 2018.

Figure R4d shows the wind speeds at the two stations are quite different due to the control of local topography. To investigate the sensitivity of freezing-induced water redistribution to wind speed, we show the total water content and water table fluctuations induced by freezing under two

different wind speeds when all other climatic conditions are the same (Figure R5). Although the total water content in the shallow part of the frozen zone is impacted by the wind speed, the total water content in the deep of the frozen zone and in the unfrozen zone, and the water table fluctuations are seldom impacted by wind speed. Therefore, we believe that the uncertainty of wind speed does not undermine our conclusion on freezing-induced groundwater redistribution.

[Figure]

Figure R5 The evolution of simulated total water content from 20 NOV 2017 to 1 APR 1 2018 based on the wind speeds of the Otok meteorological station (a) and the DK2 meteorological station (b). Note that all other climatic conditions (air temperature, relative humidity, solar radiation and snowfall) are based on the Otok meteorological station.

4. Zheng et al. (JHM, 2015, 2017) have also reported the accuracy of 5TM sensors especially in measuring unfrozen liquid water content.

Zheng, D., R. van der Velde, Z. Su, J. Wen, X. Wang, and K. Yang (2017), Evaluation of Noah Frozen Soil Parameterization for Application to a Tibetan Meadow Ecosystem, Journal of Hydrometeorology, 18(6), 1749-1763.

Zheng, D., R. van der Velde, Z. Su, X. Wang, J. Wen, M. J. Booij, A. Y. Hoekstra, and Y. Chen (2015), Augmentations to the Noah model physics for application to the Yellow River source area. Part I: Soil water flow, Journal of Hydrometeorology, 16(6), 2659-2676.

Response: Thanks for your recommendations. Zheng et al. (2015) used the 5TM sensors to measure the soil water content in the unfrozen condition, and Zheng et al. (2017) used the 5TM sensors to measure the liquid water content in the frozen soil. In Zheng et al. (2017), it was also pointed out that the accuracy of 5TM sensors is 0.02 cm$^3$/cm$^3$. Therefore, we cited Zheng et al. (2015) and Zheng et al. (2017) in our revision.

Reviewer #2

I would like to thank the authors for their substantial modifications, which address most of my concerns. I would suggest a minor revision before its publication. Please find below my specific comments for consideration.

1. bottom boundary conditions

Thank the authors for the explanations why using flux boundary conditions, not constant head/specific head for the bottom boundary condition. My concern here is why using no-flux boundary conditions. I am not sure about the amount/magnitude of the vertical flux exchange at the bottom boundary.

Response: In a completely closed system, water table would decline when groundwater is migrated toward the freezing front. In a 1D model, a no-flux lower boundary condition is the most suitable boundary condition to characterize the freezing-induced water table decline in a closed system, which had already been employed in the pioneering study of Harlan (1973).

   In our monitoring site, the water table recession is alleviated by the vertically upward component of regional groundwater inflow, the magnitude of which is controlled by the regional-scale water table undulation. Theoretically, this upward component of groundwater flow can be characterized by a specified-flux lower boundary condition. In the 1D SHAW model, such a boundary condition can be realized by combining a no-flux lower boundary condition and a specified rate of lateral groundwater inflow near the bottom. In this way, the response of water table to freezing and regional groundwater inflow is well characterized.

2. Line 39-41: "…, water from the unfrozen zone would migrate to the freezing front" suggest to be "…, water fluxes migrate from the unfrozen zone to the freezing front" or something similar.

Response: Thanks for the suggestion. We have changed the sentence into "As a result of cryosuction generated by soil freezing, (Williams and Smith, 1989;Hohmann, 1997;Yu et al., 2018), water migrates from the unfrozen zone to the freezing front."

3. Line 121: "…, we performed site-specific calibration by comparing the liquid water content measured by the 5TM sensors and by the gravimetric method." Feel awkward, please rephrase.

Response: We have changed the sentence into "we performed site-specific calibration by establishing the relationship between the liquid water content measured by the 5TM sensors and that measured by the gravimetric method."

4. Table 1 and Table 2 should better be merged into one table.

Response: Thanks for your suggestion. We have merged the two tables.

5. Figure 8: It is better to add the scenario name with different subplots, according to Table 3.

Response: Thanks for your suggestion. We have added the scenario names on the subplots.

6. This work can benefit from the English edits.

Response: Thanks for your suggestion. The language of the manuscript has been improved.

References:

Daniel, J. A., and Staricka, J. A.: Frozen Soil Impact on Grounf Water‑Surface Water Interaction, JAWRA Journal of the American Water Resources Association, 36, 151-160, 10.1111/j.1752-1688.2000.tb04256.x, 2000.

Drescher, W. J.: Some effects of preciptiation on ground water in Wisconsin, Wisconsin Geological Survey, 1955.

Evans, S. G., and Ge, S.: Contrasting hydrogeologic responses to warming in permafrost and seasonally frozen ground hillslopes, Geophysical Research Letters, 44, 1803-1813, 10.1002/2016gl072009, 2017.

Fan, Y., Li, H., and Miguez-Macho, G.: Global patterns of groundwater table depth, Science, 339, 940-943, 10.1126/science.1229881, 2013.

Harlan, R.: Analysis of coupled heat‑fluid transport in partially frozen soil, Water Resources Research, 9, 1314-1323, 10.1029/WR009i005p01314, 1973.

Hohmann, M.: Soil freezing — the concept of soil water potential. State of the art, Cold Regions Science & Technology, 25, 101-110, 10.1016/S0165-232X(96)00019-5, 1997.

Ireson, A. M., van der Kamp, G., Ferguson, G., Nachshon, U., and Wheater, H. S.: Hydrogeological processes in seasonally frozen northern latitudes: understanding, gaps and challenges, Hydrogeology Journal, 21, 53-66, 10.1007/s10040-012-0916-5, 2013.

Schneider, R.: Correlation of ground-water levels and air temperatures in the winter and spring in Minnesota: US Geol, Survey Water-Supply Paper, 1962, 1961.

Stähli, M., Jansson, P.-E., and Lundin, L.-C.: Soil moisture redistribution and infiltration in frozen sandy soils, Water Resources Research, 35, 95-103, 10.1029/1998wr900045, 1999.

van der Kamp, G., Hayashi, M., and Gallén, D.: Comparing the hydrology of grassed and cultivated catchments in the semi-arid Canadian prairies, Hydrological Processes, 17, 559-575, 10.1002/hyp.1157, 2003.

Vinnikov, K. Y., Robock, A., Speranskaya, N. A., and Schlosser, C. A.: Scales of temporal and spatial variability of midlatitude soil moisture, Journal of Geophysical Research: Atmospheres, 101, 7163-7174, 10.1029/95JD02753, 1996.

Williams, P., and Smith, M.: The frozen earth: fundamentals of geocryology, Cambridge University Press, 1989.

Wu, M., Huang, J., Wu, J., Tan, X., and Jansson, P.-E.: Experimental study on evaporation from seasonally frozen soils under various water, solute and groundwater conditions in Inner Mongolia, China, Journal of Hydrology, 535, 46-53, 10.1016/j.jhydrol.2016.01.050, 2016.

Yu, L., Zeng, Y., Wen, J., and Su, Z.: Liquid‐Vapor‐Air Flow in the Frozen Soil, Journal of Geophysical Research: Atmospheres, 123, 7393-7415, info:doi/10.1029/2018JD028502, 2018.

Zhang, T., Barry, R., Knowles, K., Ling, F., and Armstrong, R.: Distribution of seasonally and perennially frozen ground in the Northern Hemisphere, Proceedings of the 8th International Conference on Permafrost, 2003, 1289-1294,

Zhang, Z., Wang, W., Gong, C., Wang, Z., Duan, L., Yeh, T. c. J., and Yu, P.: Evaporation from seasonally frozen bare and vegetated ground at various groundwater table depths in the Ordos Basin, Northwest China, Hydrological Processes, 10.1002/hyp.13404, 2019.

Zheng, D., Velde, R., Su, Z., Wang, X., and Chen, Y.: Augmentations to the Noah Model Physics for Application to the Yellow River Source Area. Part I: Soil Water Flow, Journal of Hydrometeorology, 16, 2659–2676, 10.1175/JHM-D-14-0198.1, 2015.

Zheng, D., Rogier, V., Su, Z., Wen, J., Wang, X., and Yang, K.: Evaluation of Noah Frozen Soil Parameterization for Application to a Tibetan Meadow Ecosystem, Journal of Hydrometeorology, 18, 1749-1763, 10.1175/JHM-D-16-0199.1, 2017.